# Differential regulation of the proteome and phosphoproteome along the dorso-ventral axis of the early *Drosophila* embryo

Juan Manuel Gomez[1,2], Hendrik Nolte[3†], Elisabeth Vogelsang[2,4‡], Bipasha Dey[5], Michiko Takeda[5], Girolamo Giudice[6§], Miriam Faxel[7], Theresa Haunold[1], Alina Cepraga[1], Robert P Zinzen[7], Marcus Krüger[3], Evangelia Petsalaki[6], Yu-Chiun Wang[5], Maria Leptin[1,2]*

[1]Directors's Research and Developmental Biology Unit, European Molecular Biology Laboratory, Heidelberg, Germany; [2]Institute of Genetics, University of Cologne, Cologne, Germany; [3]Institute of Genetics, CECAD Research Center, Cologne, Germany; [4]Molecular Cell Biology, Anatomy, University Hospital Cologne, University of Cologne, Cologne, Germany; [5]RIKEN Center for Biosystems Dynamics Research, Kobe, Japan; [6]European Molecular Biology Laboratory, European Bioinformatics Institute (EMBL-EBI), Wellcome Genome Campus, Hinxton, United Kingdom; [7]Max Delbrück Center for Molecular Medicine, Berlin, Germany

**\*For correspondence:** mleptin@uni-koeln.de

**Present address:** [†]Max-Planck-Institute for Biology of Ageing, Department of Mitochondrial Proteostasis, Cologne, Germany; [‡]Molecular Cell Biology, Anatomy, University Hospital Cologne, University of Cologne, Cologne, Germany; [§]Wellcome Sanger Institute, Wellcome Genome Campus, Cambridge, United Kingdom

**Competing interest:** The authors declare that no competing interests exist.

**Abstract** The initially homogeneous epithelium of the early *Drosophila* embryo differentiates into regional subpopulations with different behaviours and physical properties that are needed for morphogenesis. The factors at top of the genetic hierarchy that control these behaviours are known, but many of their targets are not. To understand how proteins work together to mediate differential cellular activities, we studied in an unbiased manner the proteomes and phosphoproteomes of the three main cell populations along the dorso-ventral axis during gastrulation using mutant embryos that represent the different populations. We detected 6111 protein groups and 6259 phosphosites of which 3398 and 3433 were differentially regulated, respectively. The changes in phosphosite abundance did not correlate with changes in host protein abundance, showing phosphorylation to be a regulatory step during gastrulation. Hierarchical clustering of protein groups and phosphosites identified clusters that contain known fate determinants such as Doc1, Sog, Snail, and Twist. The recovery of the appropriate known marker proteins in each of the different mutants we used validated the approach, but also revealed that two mutations that both interfere with the dorsal fate pathway, *Toll*[10B] and *serpin27a*[ex] do this in very different manners. Diffused network analyses within each cluster point to microtubule components as one of the main groups of regulated proteins. Functional studies on the role of microtubules provide the proof of principle that microtubules have different functions in different domains along the DV axis of the embryo.

## Editor's evaluation

This valuable study investigated the changes in the proteome or phosphoproteome during dorso-ventral axis specification in the Drosophila embryo. The mutants used to enrich for lateral, ventral, or dorsal regions represent a solid approach, and the large datasets together with subsequent modeling experiments are compelling. This paper provides an excellent resource for the Drosophila community.

## Introduction

Morphogenesis is the developmental process that creates the three-dimensional morphology of tissues. The first morphogenetic event in metazoans is gastrulation, in which an epithelium gives rise to the germ layers from which all adult tissues derive. *Drosophila* gastrulation is probably one of the best studied embryo-scale morphogenetic processes: it is initiated by the formation of a ventral furrow that leads to the internalization of the mesoderm. The internalization of the mesoderm causes the ventral displacement of the neuroectoderm, the ectodermal cell population on the lateral side of the embryo, in the absence of particular cell shape behaviors. Finally, this ventral displacement of the neuroectoderm is accommodated by the stretching of dorsal ectodermal cells (*Leptin and Grunewald, 1990*; *Rauzi et al., 2015*). Therefore, the behavior of these cell populations can be used to study the connection between cell fate and cell shape regulation.

The behavior of a cell is determined by the identity and the state of the proteins within the cell, and by the networks through which these proteins interact. The first step to fill the gap between cell fate and cell shape behavior is to understand how the embryonic cell populations differ in their biochemical composition. Most of the cellular components pre-exist in the egg, having been provided maternally during oogenesis either as RNA or as protein. With the exception of the determinants for anterior-posterior (AP) and dorso-ventral (DV) patterning most of these proteins are distributed throughout the early embryo. As differentiation proceeds, they may be acted upon in a region-specific manner (*Gilmour et al., 2017*). For example, adherens junctions and the actomyosin meshwork are dramatically remodeled in ventral cells (*Rauzi et al., 2015*; *Kölsch et al., 2007*).

The mechanism by which the differentiation of embryonic cell populations is controlled is understood in great depth, largely through the study of mutants. Briefly, a gradient of the transcription factor Dorsal with its high point in nuclei on the ventral side is triggered by a graded extracellular signal that is transmitted through the transmembrane receptor Toll (*Moussian and Roth, 2005*; *Roth et al., 1989*). We use for our work here mutations in three genes that control dorso-ventral fates, *Toll, serpin27A* and *gastrulation defective*. Female flies that are homozygous for certain alleles of these mutations, or combinations of alleles, lay eggs that develop into embryos in which all cells express genes characteristic for only one domain of the normal embryo – either the ventral domain or the lateral or the dorsal domain – and to which we refer here as ventralized, lateralized, or dorsalized.

The transcription factors and signaling cascades set up by DV patterning and their downstream target proteins then act upon some of the maternally provided proteins in a region-specific manner. Among protein-level post-translational modifications, phosphorylation is fast and reversible and plays key roles during early embryogenesis: from regulating elements in the Toll and Dpp pathways, to the activation of the Rho Pathway within the mesoderm (*Moussian and Roth, 2005*; *Martin, 2020*). Therefore, phosphorylation is likely to be at least one way of also regulating cell behaviors along the dorso-ventral axis in a cost-effective and timely manner.

Differences between embryonic cell populations along the DV axis have been studied with transcriptomic and proteomic methods (*Casal and Leptin, 1996*; *Biemar et al., 2006*; *Gong et al., 2004*; *Stathopoulos et al., 2002*) but with limited depth and temporal resolution. Studies looking at changes over time identified proteins that appear during the maternal to zygotic transition (*Fabre et al., 2016*; *Gouw et al., 2009*) and later in embryogenesis (*Sopko et al., 2014*; *Casas-Vila et al., 2017*), but had no spatial or cell type specificity. None of these studies addressed the region-specific post-translational regulation of proteins.

To identify missing links in the pathways from known cell fate determining factors and region-specific cell behaviors, we analyzed the proteomes and the phosphoproteomes of mutants representing different cell populations along the dorso-ventral axis of the embryo. We find many proteins with differences in abundance across the populations that do not show the same differences in RNA abundance. We also find region-specific phosphorylation patterns in proteins that are ubiquitously expressed. Networks of phosphoproteins enriched in specific populations included proteasome components, RNA stress granules/P-bodies, adherens junctions associated proteins and microtubule components/associated proteins. A proof of principle test of the role of microtubules in the gastrulating embryos and revealed differential functions in the cell populations along the DV axis.

## Results

### Biological validation of dorso-ventral patterning mutants as representatives of dorso-entral cell populations in the wild type embryo

To study the proteomes and the phosphoproteomes of cell populations in the early embryo, we used mutants in which all cells in the embryo represent only one subset of the cell types present in the wild type embryo. Because we were interested in cell behaviors that affect the first step of gastrulation, which is driven by differences in cell behavior along the DV axis, we used mutants for genes of the DV patterning pathway. The embryos were derived from mothers mutant for the genes *gastrulation defective* (*gd*), *Toll* (*Tl*), or *Serpin27A* (*spn27A*). We chose those alleles that cause the strongest dorsalization, lateralization and ventralization of the dorso-ventral axis as judged by cuticle phenotypes and changes in gene expression patterns. To generate dorsalized embryos we used the $gd^9$ allele, reported to generate the strongest dorsalization without affecting the length of the embryo (*Konrad et al., 1988*; *Ponomareff et al., 2001*). Mothers transheterozygous for the hypomorphic mutations in $Tl^{rm9}$ and $Tl^{rm10}$ were used to produce lateralized embryos, in which the entire dorso-ventral axis forms neuroectoderm (*Stathopoulos et al., 2002*; *Anderson et al., 1985*; *Cowden and Levine, 2003*). Ventralized embryos were generated in two different ways using mutations in *Tl* and *spn27A*: one was to make mothers transheterozygous for the dominant *Tl* gain-of-function allele $Tl^{10B}$ (*Anderson et al., 1985*; *Schneider et al., 1991*) and deficiency *Df(3R)ro80b*, which uncovers the *Tl* locus; the other was to use mothers that were transheterozygous for $spn27A^{ex}$, an amorphic mutation (complete excision) of *spn27A* (*Ligoxygakis et al., 2003*), in combination with deficiency *Df(2L)BSC7*, which uncovers the *spn27A* locus. To confirm that the embryos produced by these mothers represented the dorsal, lateral and ventral cell populations, we analyzed the expression patterns of D-V fate determining genes (*Figure 1A*, *Figure 1—figure supplement 1B*, *Supplementary file 1*). 'Lateralized' and 'dorsalized' embryos from $Tl^{rm10}/Tl^{rm9}$ and $gd^9/gd^9$ mothers expressed neither *twist* nor *snail*, whereas ventralized embryos from $Toll^{10B}/def$ and $spn27A^{ex}/def$ mothers expressed *twist* and *snail* around their entire circumference in the trunk region (*Figure 1B*, *Supplementary file 1*). In embryos from $Tl^{rm10}/Tl^{rm9}$ mothers, *sog* expression expanded dorsally and ventrally, whereas *dpp* expression expanded ventrally (*Figure 1B*, *Supplementary file 1*). These expression patterns showed some variation and were not entirely homogeneous: ventralized embryos often had a gap in *snail* expression in a small dorsal-anterior domain around the procephalic furrow. In this region, we detected *sog* expression instead, suggesting ventralized embryos retain some cells with a neuroectodermal fate in a restricted area of the embryo (*Figure 1B*, *sog* probe).

Because we wanted to use these mutants to identify proteins that reflect or control differential cell behavior, it was important to ascertain that the cells in these mutants recapitulate faithfully the biological qualities of the corresponding cell populations in the wild type embryo (*Rauzi et al., 2015*; *Kölsch et al., 2007*), specifically of the localisation of the adherens junctions and the cortical acto-myosin meshwork. We find that, as in the mesoderm of wildtype embryos, the adherens junctions (as visualized by immunostaining for Armadillo/β-Catenin; *Figure 1C*) relocalize apically in the ventralized mutants, but remain apico-lateral in lateralized and shift slightly more basally in dorsalized mutants, again mirroring the morphology of lateral and dorsal regions of the wildtype embryo (*Figure 1C*, *Figure 1—figure supplement 1A*). Similarly, the apical actomyosin network, which we characterized in living embryos expressing a fluorescently tagged myosin light chain (sqh-mCherry, *Figure 1E and F*) forms a pulsatile apical network in ventralized embryos, whereas myosin accumulates at cell junctions in lateralized embryos, and dorsalized embryos dissolve the loose apical actomyosin of the early blastoderm (*Figure 1E and F*).

In summary, in terms of marker gene expression and cell behavior, the cells in these mutants resemble the corresponding embryonic cell populations of a wild type embryo, showing that these mutant cell populations are good sources of material to analyze the proteomic and phosphoproteomic composition of the natural cell populations at the onset of gastrulation.

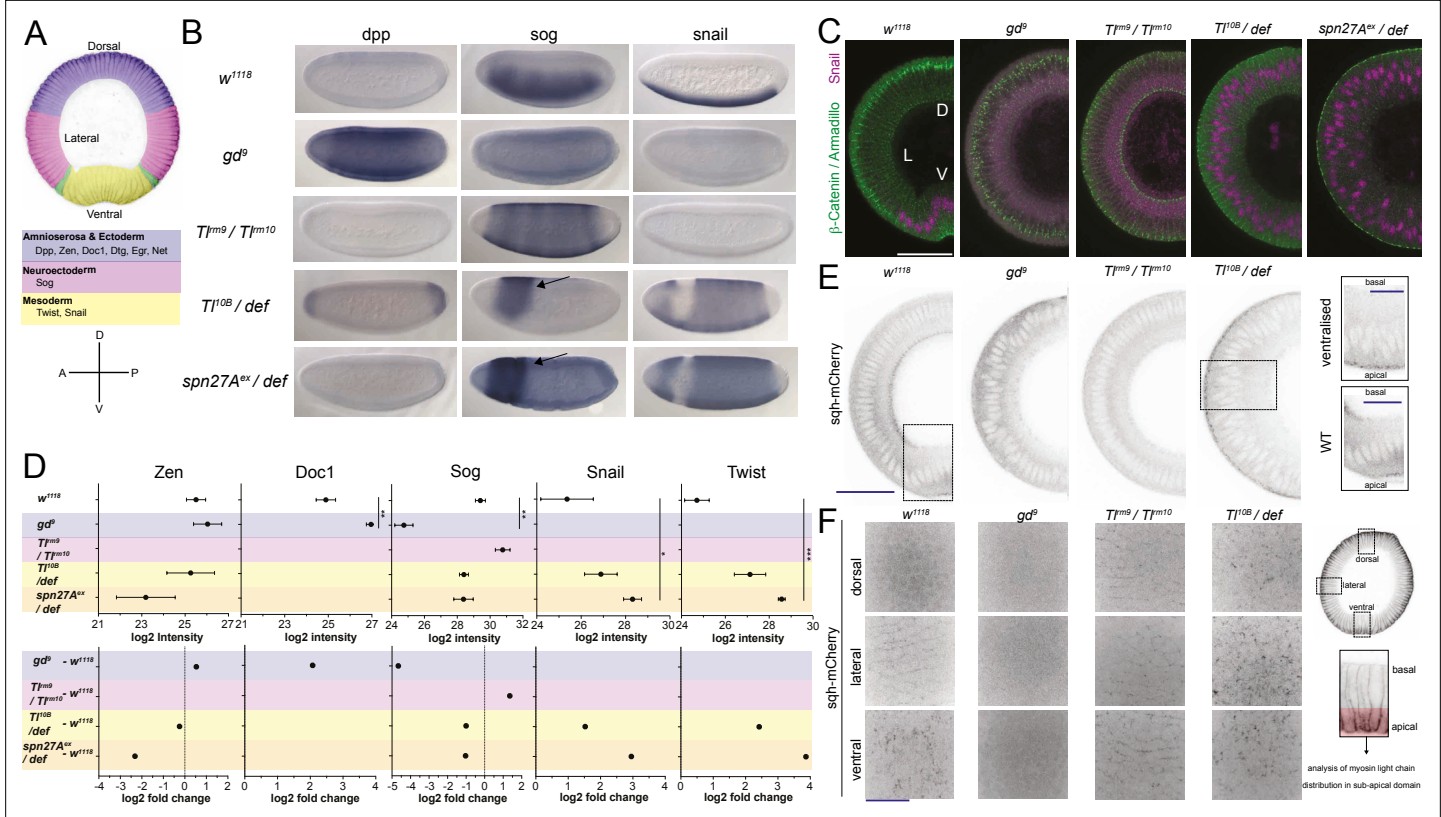

**Figure 1.** Validation of mutants as representatives of embryonic cell populations. (**A**) Top: color-coded schematic of the cell populations along the dorso-ventral axis of a *Drosophila* embryo during gastrulation: blue, 'dorsal': ectoderm and amnioserosa; magenta, 'lateral': neuroectoderm; green, mesectoderm; yellow 'ventral': mesoderm. Middle: examples of dorso-ventral fate determinants of the domains shown in the top panel. Bottom: dorso-ventral and anterior-posterior axes for reference in panel B. (**B**) RNA in situ hybridisations using probes for genes expressed in dorsal (*dpp*), lateral (*sog*) and ventral (*snail*) cell populations in wild type (*w1118*) and embryos derived from mothers mutant for dorso-ventral patterning genes (*gd9*, dorsalized, *Tlrm9/Tlrm rm10* lateralized, *Toll10B/def* and *spn27Aex/def* both ventralized). Notice the expansion of *dpp, sog* and *snail* expression in dorsalized, lateralized and ventralized embryos respectively. Arrow indicates remaining neuroectodermal polarity in ventralized embryos. (**C**) Images (confocal, max-projected) of physical cross-sections from heat-fixed embryos stained using antibodies against β-Catenin/Armadillo (green) and Snail (magenta). D is dorsal domain; L is lateral domain; V is ventral domain. Scale bar is 50 μm. (**D**) log2 intensity (top) and log2 fold change (FC, bottom) of proteins in wild type, dorsalized (blue), lateralized (magenta), and ventralized (yellow) embryos. Bars depict mean and standard error of the mean across replicates. Absence of a dot indicates the protein was not detected or log2FC calculation not feasible; absence of error bars in log2 intensity indicates protein was detected only in a single biological replicate. Dotted line indicates log2FC = 0. Mean log2 intensity values were compared using one-way ANOVA, followed by pairwise unpaired t-test comparisons (FDR corrected). Significance: * is p<0.05, ** is p<0.01, *** is p<0.001. See ***Supplementary file 2*** for ANOVA and pairwise comparison p-values. (**E**) Cross-section images (two-photon, single sections) showing Myosin Light Chain (sqh-mCherry) distribution in living wild type, dorsalized, lateralized and ventralized embryos. Insets show magnified ectopic sqh-mCherry signal distribution in wild type vs. ventralized embryos. Scale bar is 50 μm for full view and 25 μm for insets. (**F**) Images (spinning disk, max-projected) showing myosin distribution in the sub-apical domain of living wild type, dorsalized, lateralized and ventralized embryos along their dorso-ventral axis. Scale bar is 25 μm.

The online version of this article includes the following figure supplement(s) for figure 1:

**Figure supplement 1.** Proteomic and phosphoproteomic strategy in *Drosophila* embryos at the point of gastrulation.

## The proteome and the phosphoproteome of four cell populations during gastrulation

To study the proteins and the phosphosites that might be relevant for cell behavior during gastrulation, we focused on a narrow developmental time window for sample collection. We synchronized egg collections and manually collected embryos from wild type and mutant mothers aged for 165–180 minutes after egg deposition at 25 °C (Stage 6, see Methods and ***Figure 1—figure supplement 1A and B***). We analyzed their peptides and phospho-peptides with unbiased label-free quantification (LFQ) and SILAC (Stable Isotope Labeling with Amino acids in Culture ***Xu et al., 2012***; ***Sury et al., 2010***; ***Nolte et al., 2015***, ***Figure 1—figure supplement 1C–E***).

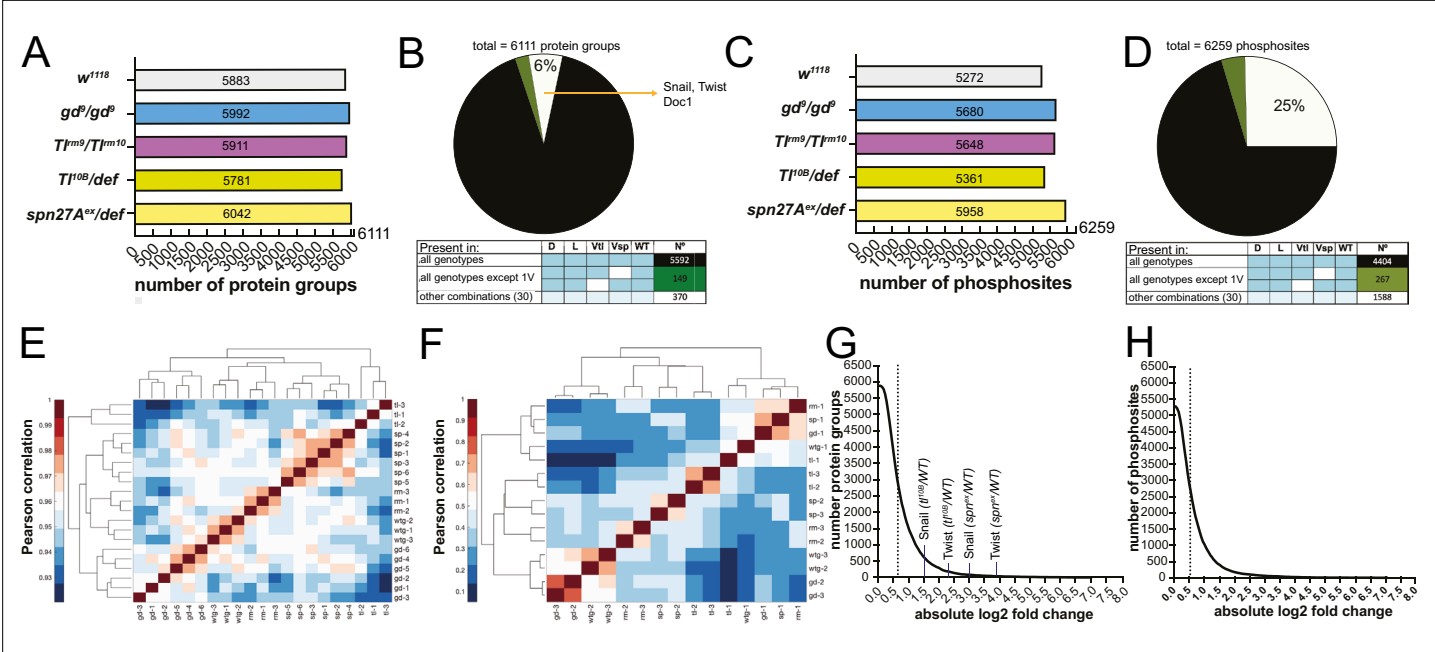

**Figure 2.** Proteomes and phosphoproteomes of wildtype and mutant embryos. (**A and C**) Number of protein groups (**A**) or phosphosites (**C**) detected in wildtype, dorsalized (*gd⁹*), lateralized (*Tl^rm9^/Tlrm^10^*), and ventralized embryos (*Toll^10B^/def* and *spn27A^ex^/def*). (**B and D**) Intersection analysis of detected protein groups (**B**) or phosphosites (**D**). Black: detected in at least 1 replicate in all genotypes; green: detected in at least 1 replicate in all genotypes except 1 ventralized condition; white: detected in at least 1 replicate in any other combination. (**E and F**) Correlation matrix between the replicates of the proteomic (**E**) and phosphoproteomic (**F**) experiments using the Pearson correlation coefficient. Protein groups and phosphosites detected in all of the replicates in all of the genotypes were used to construct the correlation matrices. Proteomic (LFQ) analyses were performed using three technical replicates, with the exception of *spn27a^ex^/def* and *gd⁹* genotypes in which we used two biological replicates with three technical replicates each, making a total of six replicates for these two genotypes. For SILAC phosphoproteomic analyzes the protein lysate from embryos of each genotype was split in three and conducted three separate analyses. (**G and H**) Distribution of the number of protein groups (**G**) or phosphosites (**H**) exceeding an absolute fold change (vs. wild type, in log2 scale). Dotted line depicts the absolute fold change corresponding to 50% of the analyzed protein groups (**G**) or phosphosites (**H**).

The online version of this article includes the following figure supplement(s) for figure 2:

**Figure supplement 1.** Proteomic validation of dorso-ventral embryonic cell populations.

In the proteomic analyses, we identified 6111 protein groups (to which, for the sake of simplicity, we will refer simply as proteins; *Supplementary file 3*) across all genotypes. A total of 5883 of these were detected in wild type embryos (*Figure 2A*), exceeding previously reported number identified by proteomic approaches in early *Drosophila* embryogenesis (*Casas-Vila et al., 2017*). Most were detected in all genotypes (*Figure 2B*). The small number (519/6111) with restricted detection included the DV fate determinants Doc1, Snail, Twist, and dMyc (*Figure 1A and B*). The phosphoproteomic analysis identified 6259 phosphosites distributed over 1847 proteins (*Figure 2C*, *Supplementary file 4*). Only 73% of phosphosites were found across all genotypes (*Figure 2D*). Twenty-eight percent of the proteins (1699/6111) and 9% of the phosphosites (573/6259) differed significantly in an ANOVA test across all five populations (wild type and four mutants; permutation-based FDR <0.1, s0=0.1).

We determined the degree of experimental variability by generating correlation matrices both for the proteome and the phosphoproteome. For the proteome, the replicates from the same genotypes clustered together (*Figure 2E*). For the phosphoproteome, the first replicate of each genotype was separated from the other two replicates (*Figure 2F*). We nevertheless kept all replicates for further analyses because it was impossible to determine the experimental source for this variation.

The enrichment for proteins or phosphosites in the mutant genotypes over the wild-type ranged from near-zero to 100 fold (*Figure 2G and H*) with about half changing by less than 1.5-fold. The fold-changes for the mesodermal fate determinants Snail (7.7 fold) and Twist (14.6 fold) measured in *spn27A^ex^/def* were the largest positive fold-changes among the DV fate determinants (*Figure 2G*).

To test if the recovered protein populations represented the cell populations in the embryo, we analyzed whether they contained known marker proteins. We first looked for the protein product of the gene that was mutated in each group of embryos. We detected both Toll and Spn27A, and each of them was reduced in abundance in the respective mutant embryos. (*Figure 2—figure supplement 1A*).

Proteins that are known to be expressed differentially along the DV axis (*Figure 1A*) were more abundant in the appropriate genotypes: Snail, Twist, Mdr49, Traf4, and CG4500 in ventralized embryos; the pro-neuroectodermal (lateral) factor Sog in lateralized embryos; pro-ectodermal (dorsal) factors Zen, Doc1, Dtg, Net, and Egr in dorsalized embryos (*Figure 1D*, *Figure 2—figure supplement 1C, D*; p values for all comparisons in all figures are summarized in *Supplementary file 2*). Known ventral-specific proteins (Snail, Twist, Mdr49, and CG4500) were more strongly upregulated in *serpin27A* embryos than in *Toll^10B*, and most dorsal-specific proteins (e.g. Egr, Zen, Sdt, Net, and Ptr) were more strongly downregulated.

We also recovered known phosphosites in proteins acting in the early embryo. This included the serine 871 phosphosite in Toll (*Zhai et al., 2008*), and serines 463, 467, and 468 in Cactus that have been shown to be phosphorylated by CKII (*Liu et al., 1997*; *Figure 2—figure supplement 1E*, *Supplementary file 2*). Toll, a known target of the Ser/Thr kinase Pelle (*Shen and Manley, 1998*), was phosphorylated on serine-871, and this phosphosite was more abundant in ventralized embryos (*Figure 2—figure supplement 1B*, *Supplementary file 2*). Phosphosites in proteins associated with the Rho pathway will be discussed below. In summary, the proteomic and phosphoproteomic screen correctly identified known and differentially expressed proteins and phosphosites.

## A linear model for quantitative interpretation of the proteomes

Our knowledge of the genetics of the dorso-ventral patterning system gives us a biological criterion that we can use to analyze the data in a stringent manner. We know that region-specific protein sets should change in concert in a well-controlled manner in all of the mutants. Rather than simply looking for individual pair-wise changes, we can, and must, therefore impose this as an additional criterion in determining any potential proteins of interest: each protein must change in a manner that 'makes sense' genetically.

The assumption that each mutant represents a defined region of the embryo makes a simple prediction for the expected outcome of the measurements: if one adds up the quantities of protein found in the mutants representing the ventral, lateral and dorsal region (normalized to the fraction of the embryo the corresponding region occupies), the sum should equal the amount of protein in the wildtype. For example, the transcription factor Snail is expressed only in the prospective meso-derm (ventral domain) in the wildtype embryo, but practically in all cells of ventralized embryos, and nowhere in lateralized and dorsalized embryos (*Figure 3—figure supplement 1A*). This is also reflected correctly in the proteomes: Snail is absent in the dorsalized and lateralized proteomes, and its level is higher in the proteomes from the ventralized embryos (*Figure 1D*, *Supplementary file 2*). Thus, Snail shows an ideal behavior in each of the DV mutant genotypes because it recapitulates the expression of Snail in the corresponding domains of a wild type embryo.

We developed a 'linear model' that is based on this additional genetic criterion, which we then used to evaluate simultaneously all mutant proteomes. We calculated for each protein the sum of its normalized quantities in the mutants and compared that sum to its abundance in the wild-type embryo. In the absence of experimental measurements for the sizes of each of the areas in the embryo (except for the mesoderm), we determined in an analytical manner (see Methods: Development of a linear model) the optimal values for the proportions occupied by the dorsal and lateral populations in the wildtype embryo, and used these to calculate the 'theoretical' wildtype value $^t\text{wt}_{\text{ProtX}}$ for each protein:

$$^t\text{wt}_{\text{ProtX}} = \mathbf{0.4D + 0.4L + 0.2V}$$

where D, L, and V are the measured abundance in the three mutant populations.

The deviation for each protein from the experimentally measured wiltype amount $^m\text{wt}_{\text{ProtX}}$ is the ratio $^t\text{wt}_{\text{ProtX}}/ {}^m\text{wt}_{\text{ProtX}}$. When we apply this analysis to one of the marker proteins, Snail, we arrive at a deviation value of 1.07 in the case where *Toll^10B* is used to represent 'ventral'. This shows that for this protein, the mutants represent the regional distribution in the wildtype very well. If we do the

calculation with *spn27A* as the ventral population the deviation value for Snail is 2.91, which indicates that this mutant genotype may over-represent the ventral population.

For our further calculations, we use the log2 of this ratio, i.e. $\text{Deviation}_{\text{ProtX}} = \log 2\ (^{\text{t}}\text{wt}_{\text{ProtX}}/^{\text{m}}\text{wt}_{\text{ProtX}})$. We found that the majority of the proteins had a deviation around zero, that is the calculated value corresponds to the measured value in the wildtype (*Figure 3A*). This would in fact be expected for any protein that is expressed ubiquitously in the wild type (such as the non-regulated maternal proteins) and should therefore be present in equal amounts in all genotypes. But even the proteins that show significant differences between at least two mutant conditions, that is the ANOVA significant subset, also fall into the range between –0.5 and +0.5, that is less than 1.4-fold deviation (*Figure 3A*, *Supplementary file 6*). This shows that the majority of proteins fit the linear model, which in turn indicates that the mutant values are good representations of protein abundance in the corresponding domains of a wild type embryo. The proteins with the most extreme deviations (more than twofold) did not come from any well-defined class of proteins, but represented a wide range of ontologies (*Figure 3—figure supplement 1D*, *Supplementary file 13*).

## Hierarchical clustering strategy and emerging regulation categories

To find the proteins that function in a tissue-specific manner during gastrulation we sorted the proteins into sets that change in concert in all of the mutants in the predicted, 'correct' manner, again using the assumptions that underlie this study, i.e. that the changes in the different mutants would be expected to correlate with each other in logical ways, as described above.

Rather than focusing only on the proteins that the ANOVA had shown as significantly modulated, we included in this analysis all proteins that were detectable in the wildtype (5883/6111), even if they were undetectable in one or more mutant populations. This allows us to include the important group of proteins that show a 'perfect' behavior, like Twist, Snail, or Doc1, in that they are undetectable in the mutants that correspond to the regions in the normal embryo where these genes are not expressed.

We used hierarchical clustering to identify the sets of proteins that change in the mutants in the same manner. For this analysis, we ignored the quantitative extent of the changes in the mutants versus the wildtype, and only focused on the direction of change if a threshold of |0.5 log2 fold change| is exceeded (see Methods). We clustered the set of 3398/6111 proteins which excluded those proteins for which the changes between the mutants and the wildtype were either all in the same direction or below the threshold.

Based on known gene expression patterns along the DV axis in the wildtype one would expect six clusters (*Figure 3C*): expression restricted to ventral (*snail*), lateral (*sog*), or dorsal (*dpp*), or expression across two domains, that is dorsal and lateral (*grh or std*), dorsal and ventral (*ama*) or lateral and ventral (*neur*). However, in addition to these clusters (marked as 1/D, 2 /L, 5 /V, 6 /DL, 9/DV and 12/LV in *Figure 3B and C*, *Supplementary file 7*) the clustering yielded a further eight clusters (*Figure 3B and C*). This results from the surprising difference between the two ventralising genotypes (*Figure 3E*). A large number of proteins change in abundance in one but not the other mutant.

Most of the marker proteins were found in their proper predicted classes (*Figure 3D*). Among those allocated to clusters where the two ventral mutants differed in their behavior, there was no general rule as to which of the two ventral mutants represented the correct value. For example, both Heartless and Net are expressed in the mesoderm and also on the dorsal side of the embryo, but Heartless was seen with increased abundance only in *serpin27A* embryos, and Net only in *Toll^{10B}* embryos (*Figure 3D*). Similarly, for genes that are excluded from the mesoderm, that is expressed in dorsal and lateral regions, some scored as present in lower abundance in *serpin27A* (e.g. crb), whereas others were reduced in *Toll^{10B}* (eg. numb). We will return to the difference between *Toll^{10B}* and *spn27A* below.

## Comparison of RNA and protein expression patterns

Protein levels can be regulated post-translationally, and RNA and protein expression levels do not necessarily correlate strongly during development (*Becker et al., 2018*). However, the regional distribution of proteins in the early *Drosophila* embryo is thought to be achieved mainly through transcriptional regulation (*Lasko, 2020*; *Ing-Simmons et al., 2021*). We therefore investigated how well the proteomes reflected known dorso-ventral modulation of gene expression.

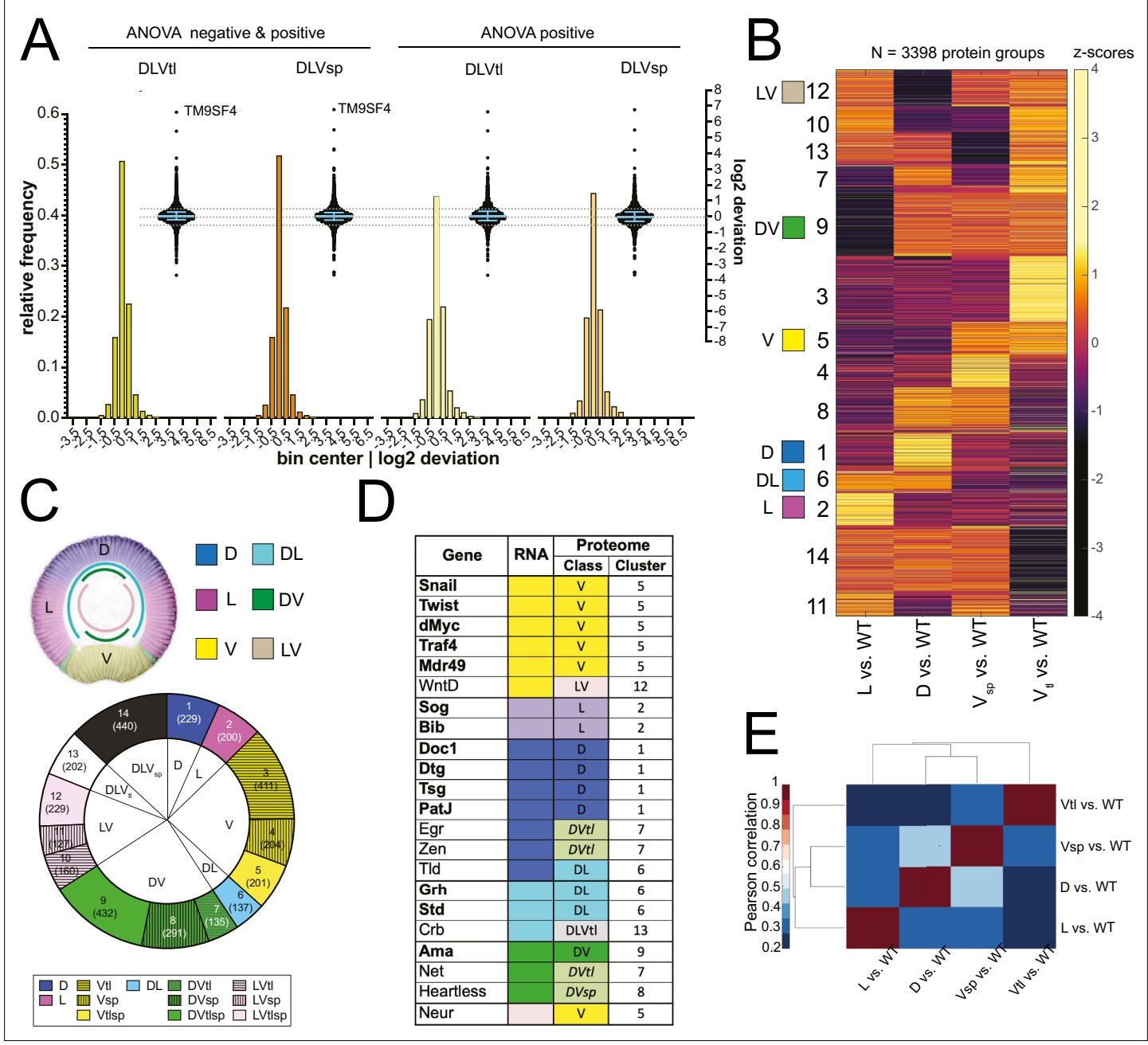

**Figure 3.** Analysis of the proteomes. (**A**) Two different representations, a histogram and a swarm plot, of the deviation parameter (in log2 scale) calculated for each of the two ventralized genotypes (D: dorsalized, L: lateralized, Vtl: *Toll^{10B}/def*, Vsp: *spn27A^{ex}/def*). This was done once for all proteins present in all genotypes, and once only for those that were ANOVA positive. In the swarm plot, each dot represents a protein. The y-axis for the histograms is shown on the left, for swarm plots on the right. Blue bars show the median with interquartile range (IQR). The median is close to zero and the IQRs range from –0.18 to 0.3. The dotted line indicates the 0.5 and –0.5 deviation in swarm plots. Histograms and swarm plots were assembled with proteins detected in all genotypes. (**B**) Clustergram of the hierarchical clustering (dendrograms not shown) of 3398 filtered proteins. Z-scores were calculated using the thresholded fold changes between DV mutants and wild type. Numbers identify the different clusters for reference across panels and figures. Coloured boxes indicate the DV clusters that show consistent behavior for the two ventralising genotypes (for color-coding see panel C). (**C**) Top: Regulation categories emerging from hierarchical clustering: D (blue), increased abundance in dorsal domain; L (magenta), increased abundance in lateral domain; V (yellow), increased abundance in ventral domain; DL (cyan) increased abundance in dorsal and lateral domains; DV (green), increased abundance in dorsal and ventral domains; LV (pink) increased abundance in lateral and ventral domains. Bottom: Pie chart showing the number of protein groups (in brackets) allocated to each cluster in (**B**), grouped by their regulation category. (**D**) Genes with restricted dorso-ventral expression with their reported RNA expression pattern, their allocation to proteome clusters (numbers refer to clusters in panel B and C), and their regulation category

*Figure 3 continued on next page*

*Figure 3 continued*

along the dorso-ventral axis. (**E**) Correlation matrices using Pearson correlation coefficient between the fold changes of each DV mutant vs. wild type in the proteome experiment.

The online version of this article includes the following figure supplement(s) for figure 3:

**Figure supplement 1.** Proteomic validation of dorso-ventral embryonic cell populations.

We first looked for genes whose RNA expression patterns are reported in the BDGP in situ database (https://insitu.fruitfly.org/cgi-bin/ex/insitu.pl; *Hammonds et al., 2013*; *Tomancak et al., 2002*; *Tomancak et al., 2007*) to compare those with ventral expression in this data set and ours. We extracted all those genes that carry the labels 'mesoderm', 'trunk mesoderm', or 'head mesoderm' in BDGP (which are not mutually exclusive). A total of 107 of the resulting set of 109 genes had their proteins detected in our analyses, and 71 had been allocated to one of the DV clusters. Sixty were found in clusters that were fully or partially consistent with the reported RNA pattern (*Supplementary file 8*). Of the 11 proteins among these 71 that show consistent mesodermal upregulation in both ventralizing mutants (DV cluster 5), all are reported as ventrally expressed in BDGP.

There is also a database representing an atlas of differential gene expression at single cell resolution for precisely the time window of early gastrulation (*Karaiskos et al., 2017*) against which we compared the proteomes to regional RNA expression (*Figure 4A*). Filtering out ubiquitously expressed genes left 8924 differentially expressed genes of which 3086 coded for 3120 proteins in our clustered proteome dataset (*Figure 4B*).

We first sorted these 3086 genes according to their expression patterns into the categories used above (D, V, L, DV, DL, LV) by virtue of similarity in their expression to six reference genes (*Figure 4A*, *Figure 4—figure supplement 1A-C*). In a second step, we excluded those that showed only spurious differences in expression along the DV axis, ending up with 155 genes with clear DV differences forming six DV RNA reference sets (*Figure 4C*, *Figure 4—figure supplement 1B-D*, *Supplementary file 9*).

We then compared the proteins in our 14 clusters against these six RNA reference sets. We asked for each protein which RNA reference set contained its corresponding gene. Theoretically, if both classifications, that is the RNA reference set and the proteomes, were perfectly correct, then genes from a protein cluster should be included only in the corresponding RNA reference set. We found that the majority of RNAs had proteins in partially or fully matching clusters of the proteomes (*Figure 4D*, *Figure 4—figure supplement 1E*). For example, nine of the thirteen proteins in cluster 5 (ventral-consistent) found their gene in the *'twist'* similarity reference group (ventral; a perfect match: white in pie charts: *Figure 4D*, *Figure 4—figure supplement 1E*). The next best matches (e.g. ventral plus lateral, instead of only ventral; a partial match, gray) were often also highly represented: three of the four remaining cluster 5 proteins found their gene in the *'neur'* similarity reference group (lateral +ventral).

Thus, the majority of proteins had perfect or partial matches with the RNA expression, showing that two independent measurements of regional expression patterns arrive at the same allocation. This confirms in an unbiased manner that the hierarchical clustering successfully sorted the proteomes in the correct manner, further supporting the initial assumption that the mutant populations were representative of specific regions in the embryo.

## Different effects of the *Toll*[10B] and *spn27A* mutations on dorsal gene expression

The difference between the results for the *Toll*[10B] and *spn27A* embryos was an unexpected and potentially biologically interesting discovery. We investigated whether the matching of the protein distributions to their RNA expression patterns could give us further biological insights.

We find that for those clusters in which *Toll*[10B] and *spn27A* agree, a larger proportion of proteins is allocated to the correct RNA reference set than in the clusters in which *Toll*[10B] and *spn27A* differ (*Figure 4D and E*, *Figure 4—figure supplement 1E*). The ventral cluster 5, in which *Toll*[10B] and *spn27A* agreed, included Snail, Twist and other genes expressed in the mesoderm (*Figure 1D*, *Figure 2—figure supplement 1B*), such as Mdr49 (*Stathopoulos et al., 2002*), CG4500 (*Casal and Leptin, 1996*), and Traf4 (*Mathew et al., 2011*).

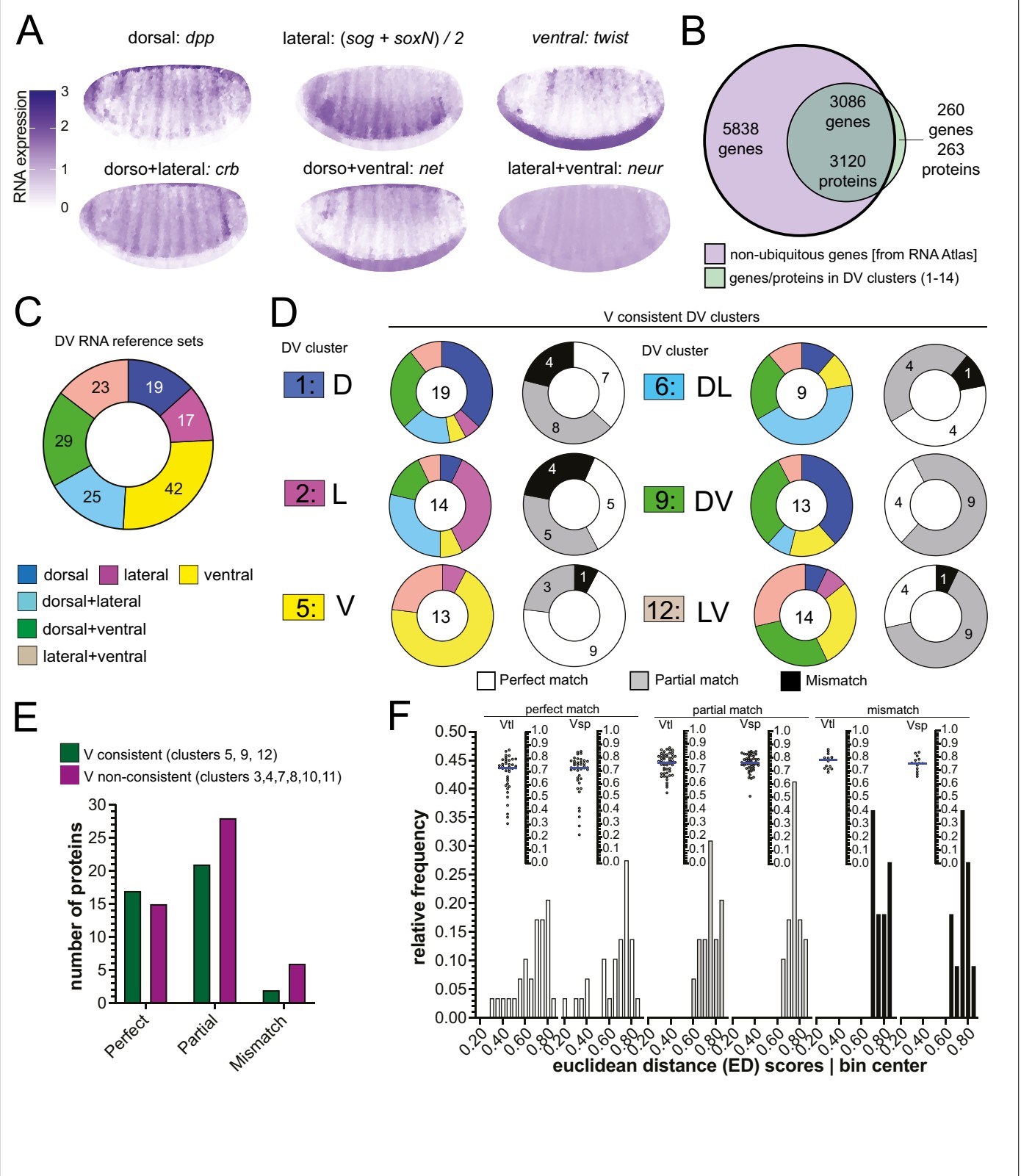

**Figure 4.** Comparison of RNA and protein expression patterns. (**A**) Expression patterns of the reference genes used for assembling the RNA reference sets. Scale bar indicates RNA expression strength. Blue depicts high RNA expression. (**B**) Venn diagram showing the intersection between the genes of the RNA atlas with non-ubiquitous expression (8924) and the genes that encode the proteins assigned to DV clusters (3383/3398 proteins were successfully matched to a FBgn gene identifier). (**C**) Number of non-ubiquitous genes allocated to each DV RNA reference set that trespassed

*Figure 4 continued*

the corresponding filter/threshold values (155/8924 genes, see Methods and Supp. **B–D**). (**D**) Matching of genes in the RNA reference sets with the proteome DV clusters in which the ventralized mutants display a consistent behavior against the wild type (DV clusters 1, 2, 5, 6, 9, and 12, see *Figure 3B and C*). Coloured pie charts represent the allocation of genes within each DV cluster to the RNA reference sets (color code as in *Figure 3*). Grayscale pie charts represent the same sets of proteins, but marked by the outcome of the comparison between RNA and protein expression inferred from clusters: white: perfect match; gray: partial, if the RNA reference included the correct match but also another region; black: mismatch, where the protein expression did not overlap with the RNA reference. Values in the center of pie charts indicate the number of genes compared. Numbers in grayscale pie charts indicate the number of proteins with a perfect (white), partial (gray) or mismatch (black). (**E**) Number of proteins in each outcome of the RNA-proteome comparison (perfect, partial, mismatch) in ventralized-consistent clusters (dark green) and ventralized non-consistent clusters (dark magenta) that belong to the same regulation categories: ventral (3,4,5), dorsal + ventral (7,8,9) and lateral + ventral (10,11,12). (**F**) Two different representations, histogram and a swarm plot of the distributions of the Euclidean distance score for proteins that had a perfect, partial or mismatching overlay with the DV RNA reference sets. The histogram and the scatter plots are shown separately for calculations using each ventralized genotype: Vtl = *Toll^{10B}/def*; Vsp = *spn27A^{ex}/def*. In the swarm plots, each dot is a protein.

The online version of this article includes the following figure supplement(s) for figure 4:

**Figure supplement 1.** Methodology and supporting data for the comparison between RNA and protein abundance.

For the proteins from the 'ventral inconsistent' clusters we found that the *Toll^{10B}* mutant differs from the *spn27A* mutant in a consistent manner. Proteins classified on the basis of being upregulated in *Toll^{10B}* (clusters 3, 7, 10, and 13) are often mismatched to genes with an ectodermal expression (dorsal and/or lateral RNA), whereas this does not occur for those classified based on their upregulation in *spn27A* (clusters 4, 8, 11, and 14, *Figure 4—figure supplement 1E*). This means that although *Toll^{10B}* mutants are strongly ventralized in terms of morphology and upregulation of ventral genes, the ectopic Toll signaling in the mutant fails to suppress all dorsal markers, which is consistent with our observation that *spn27A* mutants show a stronger reduction in dorsal-specific proteins. This confirms previous suggestions that *spn27A* mutants retain no or almost no DV polarity, whereas *Toll^{10B}* embryos retain residual polarity (*Roth et al., 1989*; *Ligoxygakis et al., 2003*). Determining the developmental source of these differences goes beyond the scope of this study, but will warrant further investigation.

## RNA-protein match versus degree of differential expression

We wondered whether there were consistent differences between those proteins that matched their RNA and those that did not. For example, a protein with large fold-changes may be more likely to match the correct RNA distribution. Because the clustering assigned proteins only on direction and not on extent of change, clusters also contain proteins with very small differences between the DV populations, even in cases where the RNA is known to show a clear difference (e.g. Traf4; *Figure 2—figure supplement 1C*).

To distinguish between strong and weak differential expression, we ranked proteins by comparing them to the most extreme protein in each cluster, that is the one that showed the greatest fold changes in the mutants over wildtype. We calculated the Euclidean distance (ED) between each protein and the most extreme (see Methods, *Supplementary file 11*). Thus, proteins with the lowest ED scores are those that are closest to the most extreme protein. We then analyzed if this score correlated with the degree to which a protein matched its RNA expression. We found that proteins from the 'matching' groups had ED-scores that were skewed towards lower values (*Figure 4F*) indicating that proteins with more extreme expression differences (low ED scores) are more likely to match the correct RNA expression pattern.

In summary, these approaches stratify our results in a useful manner: first, the DV clusters in which the two ventralized mutants behave consistently represent better the RNA expression patterns; second, proteins with strong fold-changes are more likely to represent the distribution of the corresponding RNA.

## The phosphoproteome of embryonic cell populations during gastrulation

Changes in the abundance of phosphosites may occur for two reasons: either the protein itself varies in abundance, or the protein level is constant, but the protein is differentially phosphorylated. Combinations of these cases are possible, and protein abundance may be affected by phosphorylation itself. Since we know the changes in protein abundance, we can distinguish these cases by comparing

the full proteome against the phospho-proteome (with the caveat that, for technical reasons, our measurements were done on parallel experiments rather than on the identical samples). 1765 of the phospho-proteins (96%) we identified were ones that we also found in the proteome, whereas 82 had not been detected in the proteome (*Figure 5A*). We found that most of the changes in phosphorylation were in proteins for which the level of the host protein was unchanged (black and white boxes in *Figure 5B*; 67 to 82% of the protein-phosphosite pairs). Among those for which the host protein showed differential abundance, 7–13% of their phosphosites changed in the same direction (both protein abundance and phosphorylation up, or both down), and 10–19% changed in the opposite direction.

We tested if the phospho-proteomes fitted the 'linear model' (i.e. whether the sum of the weighted mutant values corresponded to the measured values in the wildtype) and found that the majority of the phosphosites did (*Figure 5C*, *Supplementary file 6*). Among the strongly deviating phosphoproteins, we find a number of kinases with known morphogenetic functions, such as Par-1, SRC42A and nucleoside-diphosphate kinase (*awd*) (*Figure 3—figure supplement 1E*, *Supplementary file 13*).

We clustered the phosphosites using the same procedure as for the proteome. After excluding sites that were unchanged or up- or down-regulated in the same direction in all mutants, clustering the remaining 3433 phosphosites again yielded 14 DV clusters (*Figure 5D and H*, *Supplementary file 7*). The two ventralising mutants now clustered together, and the dorsalized mutant showed the most distinct behavior (*Figure 5G*, compare to *Figure 3E*).

## Emergence of differentially regulated networks of proteins and phosphoproteins along the DV cell populations

One aim of this study was to find cellular components that are differentially modified along the DV axis and that are candidates for regulating cell shape. Most likely, these cellular components are regulated by protein complexes or interacting protein networks, as already known for the regulation of actomyosin by the Rho pathway and some components of adherens junctions. Rho is activated and necessary for cell shape changes in the mesoderm, but we do not know the full set of the components of the pathway that are modulated in the mesoderm or elsewhere along the DV axis. We therefore looked at this pathway. Of 24 proteins associated with Rho signaling, we detected 21 in the wild type and at least one of the mutants (*Figure 5E*). Most, including the myosin light chain, occurred at similar levels in all genotypes, except Cofilin/Twinstar, Moesin, and Profilin/chickadee, which were more abundant in the ectodermal cell populations (clusters D and DL).

Fourteen of the 21 proteins were phosphorylated. These included the known phosphosites in myosin light chain (MLC) and Cofilin/Twinstar (*Figure 5F*, no statistical differences across genotypes for Sqh and Cofilin phosphosites, see *Supplementary file 2*), and the phosphorylation of the Cofilin/Twinstar kinase LIMK1 and phosphatase Slingshot (ssh), which were modulated in the D and DL clusters, as were RhoGEF2 and the MLC phosphatase Mbs (*Figure 5E*). In summary, we detected most of the elements of a well established pathway required for gastrulation and also identified new candidate regulation nodes within the Rho pathway.

To systematically find such networks, we used a diffusion-based algorithm (*Giudice et al., 2024*) on each of the DV clusters. The starting weight of each protein was based on either on its euclidean distance score ('ED', *Supplementary file 11*) or on the deviation from the linear model ('Dev', *Supplementary file 6 and 10*). Since these scores existed separately for the two ventralizing mutants, we also had to conduct the analyses twice in each case, that is once for each dataset. We focused our analyses only on the six DV clusters in which the ventralized mutants agree (D(1), L(2), V(5), DL(6), DV(9), and LV(12)). Overall, this resulted in 24 protein networks (ED score for each of the ventralizing mutants and deviation score for each mutant, each applied to the 6 clusters) for the proteome and 24 for the phospho-proteome. An ego network analysis (see Methods) yielded a set of 83 ontology terms in the proteome and 87 in the phospho-proteome that were significantly enriched in one or more networks. We concentrated our further analyses only on those ontology terms that were enriched in at least two of the four networks for each DV cluster and used a heatmap to represent them (*Figure 6A and B*, *Figure 6—figure supplement 1A, B*). The heatmaps illustrate that both experiments were highly enriched for cellular components associated with DNA and RNA metabolism or the regulation of gene expression. This is not unexpected for this developmental period of dynamic changes in gene expression. In agreement with this, the majority of the enriched proteins and phosphoproteins were

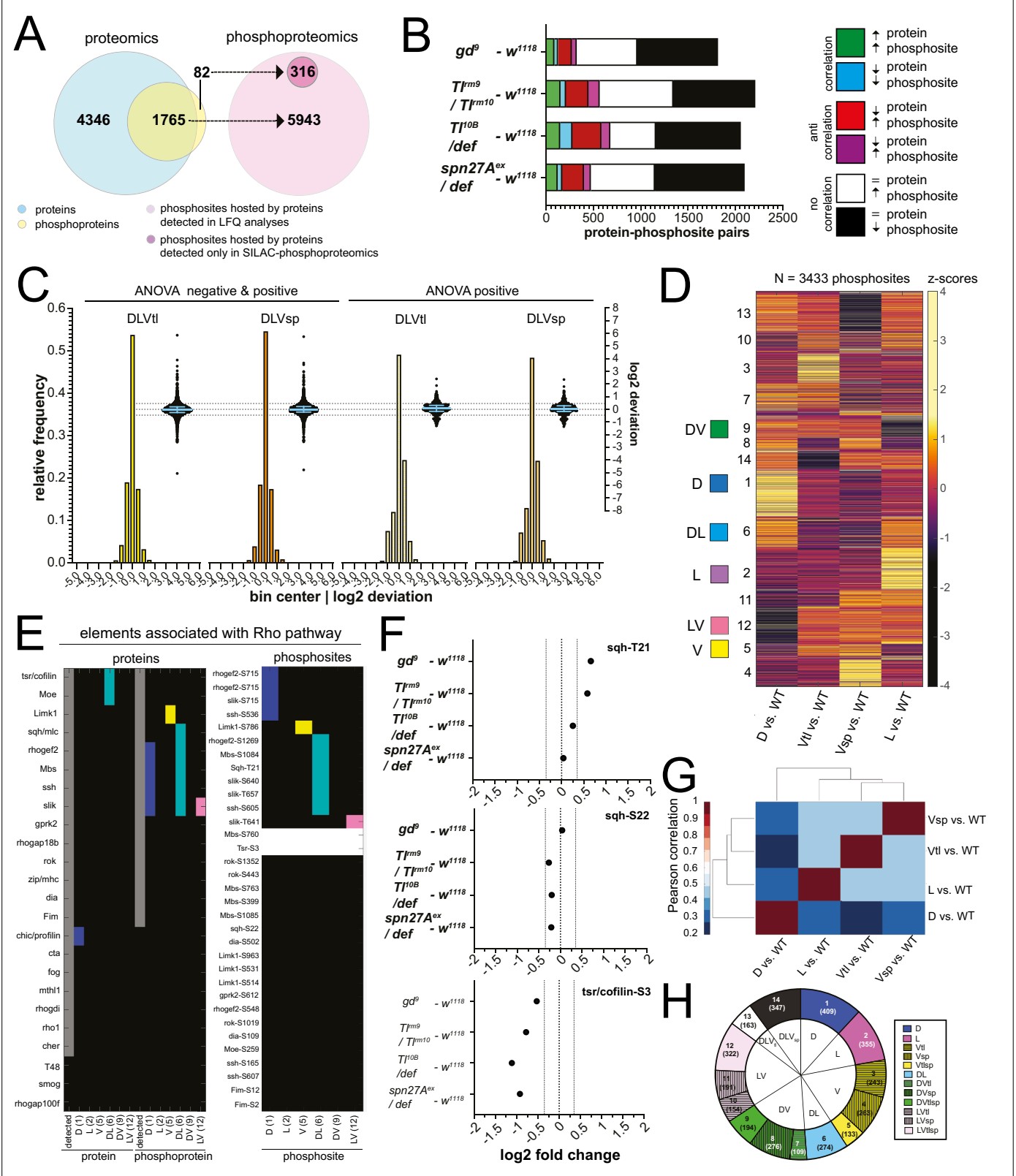

**Figure 5.** The phosphoproteomes of the mutant embryos. (**A**) Match between detected protein groups in the proteomic and phosphoproteomic experiments. Left: light blue: protein groups detected in proteomes (LFQ); overlay between light blue and yellow: protein groups detected both in proteomic and phosphoproteomic experiments; non-overlapping yellow: protein groups not detected in proteomes but detected in phosphoproteomes. Right: pink: phosphosites hosted by a protein group detected in the proteomic analyses; magenta, phosphosites that could not

*Figure 5 continued on next page*

*Figure 5 continued*

be matched to a protein group detected in the proteomes. (**B**) Correlation between the fold changes (FCs) of phosphosites and their host proteins in DV mutants vs. wild type. Correlations: FC of protein and phosphosite are both positive (green) or negative (blue). Anti-correlation: FC of protein and phosphosite have different signs (red and magenta). No correlation: protein levels are unchanged but phosphosite FC is positive (white) or negative (black). Bars represent the number of phosphosite-host protein pairs falling in each correlation category within each DV mutant vs. wild type comparison. (**C**) Two different representations, a histogram and a swarm plot, of the deviation parameter (in log2 scale) calculated for each of the two ventralized genotypes (D: dorsalized, L: lateralized, Vtl: *Toll¹⁰ᴮ/def*, Vsp: *spn27Aᵉˣ/def*). This was done once for all phosphosites present in all genotypes, and once only for those that were ANOVA positive. In the swarm plot, each dot represents a phosphosite. The y-axis for the histograms is shown on the left, for swarm plots on the right. Blue bars show the median with interquartile range (IQR). The median is close to zero and the IQRs range from –0.2387 to 0.3197. The dotted line indicates the 0.35 and –0.35 deviation in swarm plots. Histograms and swarm plots were assembled with phosphosites detected in all genotypes. (**D**) Clustergram of the hierarchical clustering of 3433 phosphosites. Z-scores were calculated using the thresholded fold changes between mutants and wild type. Coloured boxes indicate the clusters with consistent behavior for the two ventralising genotypes (for color-coding see *Figure 3D*). Numbers identify the different clusters for reference across panels, and are equivalent to the proteome (*Figure 3B and C*). (**E**) Detection and predicted regulation (DV clusters) of Rho pathway proteins and phosphoproteins (left panel) and the corresponding phosphosites (right panel). Colors mark proteins, phosphoproteins or phosphosites in DV clusters with ventralized consistent behavior(DV clusters 1, 2, 5, 6, 9, and 12). Gray boxes represent the detection of a particular protein or phosphoprotein in the wild type genotype. White boxes represent an increased or decreased abundance in all DV mutants vs. wild type. (**F**) log2 fold changes (FC) of known phosphosites in sqh: T21 (top) and S22 (center) and tsr/cofilin S3 (bottom). Dotted lines indicate log2 FC = 0, 0.35 and –0.35. Colors depict DV mutant genotypes and their corresponding comparisons against wild type: blue: dorsalized, magenta: lateralized, yellow: ventralized (*Toll¹⁰ᴮ/def* and *spn27Aᵉˣ/def*). Bars depict mean and standard error of the mean across replicates. Absence of a dot indicates the protein was not detected in a particular condition or log2 FC calculation not feasible, absence of error bars in log2 intensity indicate protein was detected in a single replicate. Dotted line indicates log2 FC = 0 and log2 FC = 0.35 (for phosphosites). (**G**) Correlation matrices using Pearson correlation coefficient between the fold changes of each mutant vs. wild type comparison in the phosphoproteome experiment. (**H**) Pie chart showing the number of phosphosites allocated to each DV class in (**C**).

characterized as nuclear ontology classes. Because of our interest in morphogenesis we focused on the cellular components that belong to cytoskeletal, cell adhesion, and vesicle trafficking categories. In the phosphoproteomes the networks enriched for cytoskeletal components were much more prevalent in the phosphoproteomes (14 of 62) than in the proteomes (3 of 63), with microtubules strongly represented (12 of 14 cellular components), in particular the alpha and beta tubulins and microtubule associated proteins (*Supplementary file 12*). Cytoskeletal proteins are often localized in the cell cortex, and we indeed find this association reflected in the results of the network analysis. The cell cortex is among the enriched components, and among the proteins in this category, we find cytoskeletal elements. For example, networks that include the actin-microtubule crosslinker Shot and the actin polymerase Profilin are enriched in the dorsal cluster; networks that include the apical polarity determinant Stardust or the Hippo pathway component Warts in the dorso-lateral cluster. A phosphoprotein network associated with adherens junctions and zonula adherens, one of which contains the junction-actin connectors Canoe and Girdin (*Houssin et al., 2015*; *Sawyer et al., 2009*) was enriched in the D cluster (*Supplementary file 12*).

In summary, we can highlight two outcomes of the network propagation analysis. First, most networks, whether derived from the proteomes or the phosphoproteomes, are enriched for cellular components associated with regulation of gene expression (transcription, epigenetic regulation, translation, protein turnover). This is a useful validation of the approach, in that it reflects the main biological process that occurs at this stage of development: giving cells in the body different developmental fates, which is achieved through setting up different gene expression programmes. Secondly, the cytoskeleton emerges as a major target of regulation in the phosphoproteome, with the most prominent component being the microtubules. This is an interesting target for further exploration in the context of gastrulation and fits well with recent results that microtubules play a role in epithelial morphogenesis (*Takeda et al., 2018*; *Ko et al., 2019*; *Booth et al., 2014*; *Pope and Harris, 2008*).

## Functional implications of networks enriched for microtubule components

We tested the biological relevance of the predicted phospho-regulation of microtubule networks. Before gastrulation, all cells have two subpopulations of MTs, which differ in their post-translational modifications: a disordered apical network of non-centrosomal MTs with short, non-aligned filaments, and an 'inverted basket' of basal-lateral MTs originating from the centrosomes and enclosing the nucleus (*Takeda et al., 2018*; *Viswanathan et al., 2019*; *Figure 7A*). The apical population contains

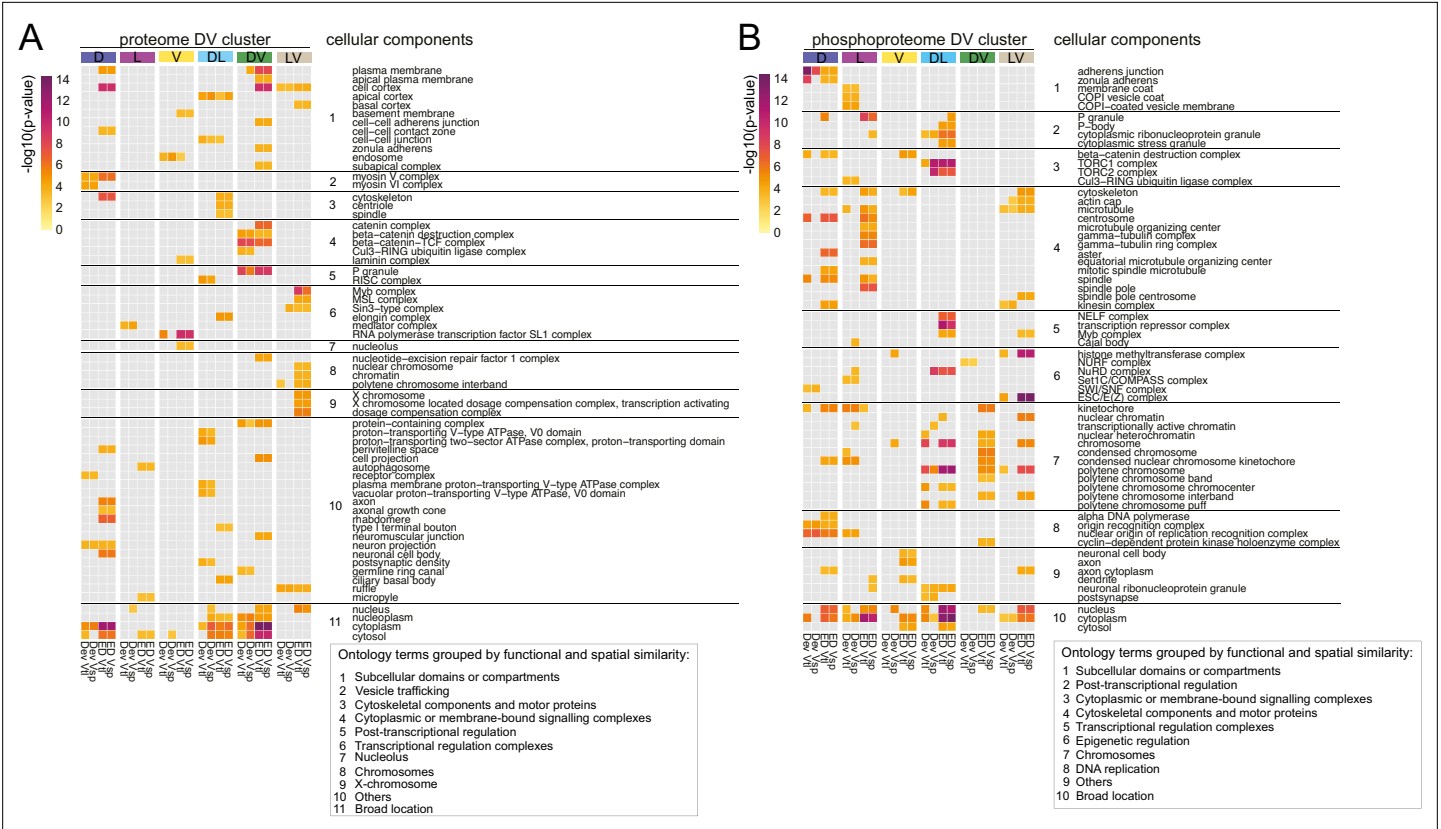

**Figure 6.** Diffused network analyses of DV proteomes and phosphoproteomes. Heatmap representations of cellular component ontology terms that were significantly enriched in at least two networks across all DV clusters and showed a consistent behavior in the ventralized genotypes (DV clusters 1, 2, 5, 6, 9, and 12). Ontology terms were grouped based on spatial and functional association. (**A**) Cellular components enriched in networks emerging from the proteome. (**B**) Cellular components enriched in networks emerging from the phosphoproteome. Calibration bar indicates the -log10(p-value) for a measure of statistical significance across ontology terms and DV clusters. For each DV class, four diffused networks were generated using the deviation ('Dev'), or the euclidean distances ('ED') to score the nodes of emerging networks. Calculations were performed independently for each score and each ventralized genotype, Vtl (*Toll^{10B}/def*) and Vsp (*spn27A^{ex}/def*).

The online version of this article includes the following figure supplement(s) for figure 6:

**Figure supplement 1.** Cellular component terms significantly enriched in diffused networks.

only dynamic MTs, marked by tyrosinated α-tubulin, whereas the inverted basket also contains stable MTs, marked acetylated α-tubulin (*Takeda et al., 2018*; *Wloga et al., 2017*; *Figure 7A*, *Figure 7— figure supplement 1A*). During gastrulation MT acetylation patterns change. In the ectoderm, MTs become increasingly acetylated but retain their original organization whereas in central mesodermal cells, the basal-lateral MTs become less acetylated (*Figure 7B*, *Figure 7—figure supplement 1A*). Some MTs in non-constricting mesodermal cells align below the apical surfaces of these cells as they extend towards the ventral midline (*Figure 7—figure supplement 1B*, arrow). These MTs are non-acetylated, but partially tyrosinated (*Figure 7—figure supplement 1A*, blue arrowhead).

We depolymerised microtubules with Colcemid and observed the ensuing cellular dynamics. Less than 1 min after the injection, most apical filamentous structures, astral MTs emanating from the centrosome, and the centrosomes themselves disappeared while the stable MTs associated with the nuclear envelope were partially retained (*Figure 7—figure supplement 1C, D*).

Colcemid treatment affected nuclear positioning and cell morphogenesis. Nuclei normally move basally for 1~2 μm in the last ~20 min of cellularization, and this failed in Colcemid-treated embryos, where the nuclei moved slightly further towards the apical cell surface (*Figure 7C–K*, *Videos 1–3*).

In normal embryos, nuclei in the constricting ventral furrow cells move a long way from the apical cell surface. In Colcemid-injected embryos, nuclear positioning was more random (*Figure 7C* and t0; 7L,7M). Ultimately, the ventral furrow failed to form (*Figure 7C*, $t_0$ +10', *Video 1*).

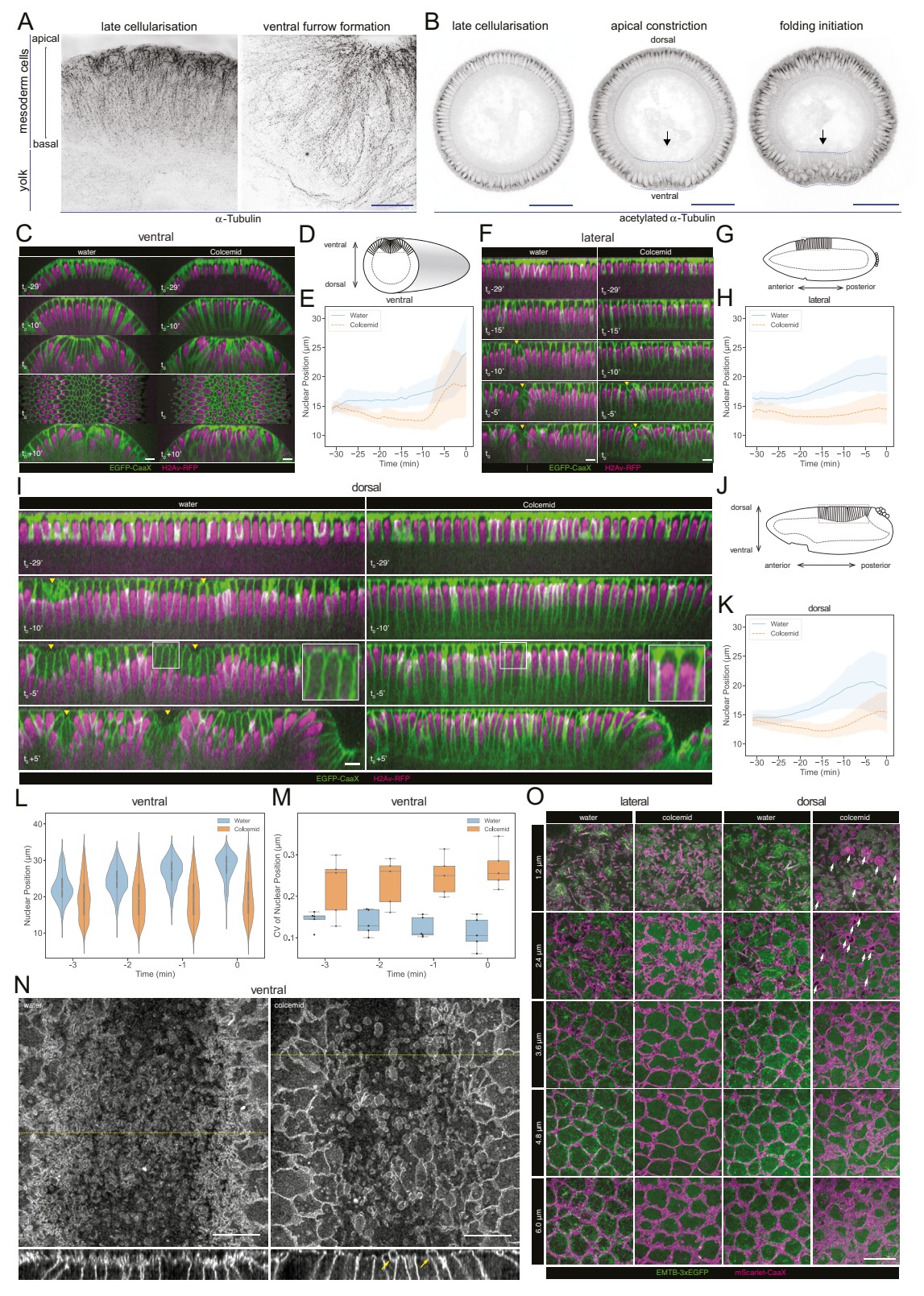

**Figure 7.** Microtubule organization and in vivo functions. (**A**) Images (OMX super-resolution microscope, max-projected) of mesoderm cells (ventral domain) using physical cross-sections from fixed embryos stained with antibodies against α-Tubulin. Left panel: onset of gastrulation, right panel: contractile mesoderm during ventral furrow formation. Scale bar is 10 μm. (**B**) Images (confocal, max-projected) of physical cross-sections from fixed embryos stained with an antibody against acetylated α-Tubulin at the onset (left panel) and during ventral furrow formation (center panel: initiation

*Figure 7 continued on next page*

*Figure 7 continued*

of gastrulation, apical constriction; right panel: mesoderm folding). Arrow indicates detection of acetylated α-Tubulin specifically in basal-lateral microtubules (inverted basket). Dotted blue line encloses mesodermal cells, in which a progressive reduction of acetylated α-tubulin is detected during ventral furrow formation. Scale bar is 50 µm. (**C, D, E**) Phenotypic effect of colcemid injection on the ventral side of the embryo during cellularization and early gastrulation. (**C**) Time-lapse series of Z re-slice (and a surface projection for the $t_0$ time point) showing cellular and tissue architecture with membrane (EGFP-CaaX) and nucleus (H2Av-RFP) labels. (**D**) A schematic drawing of tissue architecture during ventral furrow formation with a dotted rectangular box depicting the ROI of the re-slice view in panel C. (**E**) Nuclear position, that is distance from the embryo surface, as a function of time during cellularization (Water: 26-266 nulcei from 5 embryos; Colcemid: 112-350 nulcei from 5 embryos). (**F, G, H**) Same as above for the lateral side of the embryo with (**F**) time-lapse series of Z re-slice, (**G**) a schematic drawing during cephalic furrow formation and a dotted rectangular box for the ROI in panel F, and (**H**) nuclear position as a function of time (Water: 82-158 nulcei from 3 embryos; Colcemid: 100-161 nulcei from 4 embryos). (**I, J, K**) Same as above for the dorsal side of the embryo with (**I**) time-lapse series of Z re-slice, (**J**) a schematic drawing during dorsal fold formation and a dotted rectangular box for the ROI in panel E, and (**K**) nuclear position as a function of time (Water: 88-208 nulcei from 5 embryos; Colcemid: 83-211 nulcei from 7 embryos). Insets, enlarged view showing the shape of the apical dome. (**L, M**) Colcemid treatment leads to a wider distribution of nuclear positions in apically constricting VF cells during the early phase of apical constriction, shown in a violin plot for nuclear centroid position (**L**; Water: 7-125 nulcei from 5 embryos; Colcemid: 67-180 nulcei from 5 embryos) and a box plot for its coefficient of variation (**M**; Water: 5 embryos; Colcemid: 5 embryos). For all of the above, $t_0$ represents the onset of gastrulation, as defined in M&M. Yellow arrowheads: surface clefts resulting from cephalic furrow (**F**) and dorsal fold (**I**) initiation. (**N**) Apical surface projection (top row) of membrane (3xmScarlet-CaaX) and Z re-slice (bottom row; taken from the yellow dotted lines in the top row) showing enlarged membrane blebs (yellow arrows) after colcemid injection during ventral furrow apical constriction. (**O**) Apical membrane phenotypes in the lateral and dorsal cells observed at different Z positions, each with a 1.2 µm projection, visualized with membrane (3xmScarlet-CaaX) and microtubule (EMTB-3xEGFP) labels. White arrows: abnormal membrane blebs that are devoid of microtubules and observed exclusively on the dorsal side. Scale bars: 10 µm.

The online version of this article includes the following figure supplement(s) for figure 7:

**Figure supplement 1.** Microtubule analyses during gastrulation and calibration of colcemid injections.

Nuclei were also positioned apically in the neuroectoderm in Colcemid-injected embryos. The formation of the cephalic furrow was delayed by 5 min, but its progress was not affected by Colcemid-treatment (*Figure 7F–H*, *Video 2*).

Cells on the dorsal ectoderm form an apical dome with a characteristic, curved cell apex (*Figure 7I*, $t_0$-5', insets) which is abolished in Colcemid-injected embryos, supporting the model that MT-dependent force is required for apical dome formation (*Figure 7I*, $t_0$-5', insets). The dorsal epithelium forms folds which depend on the remodeling of apical MTs, but not on myosin contractility (*Rauzi et al., 2015*; *Takeda et al., 2018*; *Wang et al., 2012*) and involves the descent of the apical dome in initiating cells (*Figure 7I*, $t_0$ +5'). Dome descent does not occur in Colcemid-injected embryos, and dorsal fold formation eventually fails (*Figure 7I*, $t_0$ +5', *Video 3*), supporting the current model that microtubule forces also engage on cell shortening during Dorsal Fold Formation (*Takeda et al., 2018*; *Wang et al., 2012*).

MT depolymerization also affected the apical plasma membrane dynamics. Blebs in the apical membrane of constricting mesodermal cells (*Costa et al., 1994*) were strikingly enlarged after Colcemid injection (*Figure 7N*). Lateral and dorsal cells lacked these constriction-dependent blebs. Nevertheless, after Colcemid injection,

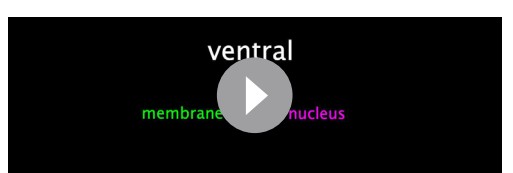

**Video 1.** Live imaging of ventral furrow formation in a representative *Drosophila* embryo (gastrulation stage 6), after water (control, left panel) or Colcemid injection (right panel). Membranes are labeled in green (EGFP-CaaX) and nuclei are labeled in magenta (H2Av-mRFP1). Scale bar is 10 µm.

https://elifesciences.org/articles/99263/figures#video1

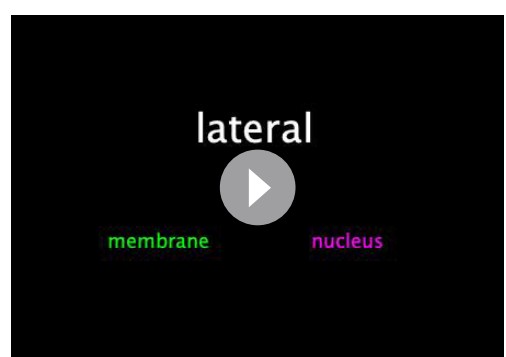

**Video 2.** Live imaging of cephalic furrow formation along the lateral side of a representative *Drosophila* embryo (gastrulation stage 6), after water (control, left panel) or Colcemid injection (right panel). Membranes are labeled in green (EGFP-CaaX) and nuclei are labeled in magenta (H2Av-mRFP1 [71]). Scale bar is 10 µm.

https://elifesciences.org/articles/99263/figures#video2

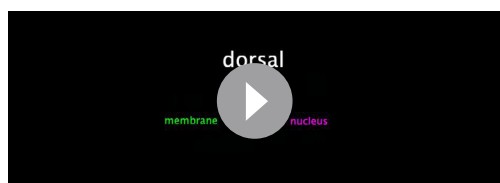

**Video 3.** Live imaging of dorsal fold formation (mid-sagittal view) of a representative *Drosophila* embryo (gastrulation stage 6), after water (control, left panel) or Colcemid injection (right panel). Membranes are labeled in green (EGFP-CaaX) and nuclei are labeled in magenta (H2Av-mRFP1 [71]). Scale bar is 10 μm.
https://elifesciences.org/articles/99263/figures#video3

they accumulate excessive, tortuous subapical membrane (*Figure 7O*; *Figard et al., 2013*; *Fabrowski et al., 2013*). We also observed a distinct class of micron-scale membrane blebs in all dorsal cells, not limited to the dorsal fold initiating cells and unrelated to myosin-dependent apical constriction (*Figure 7O*). These blebs form during mid to late cellularization, exclude MTs and are stable for minutes.

In sum, and consistent with a role for microtubules predicted by diffused network analyses, MTs are required for correct nuclear positioning and cell shape homeostasis, and have distinct functional requirements in all three types of epithelial folds during *Drosophila* gastrulation. Distinct phenotypes of the apical membrane following Colcemid injection suggest differential functionality in the maintenance of membrane-cortex attachment or the dynamics of apical membrane retrieval for MT networks residing on different sides of the embryo.

## Discussion

We have presented a large-scale study of regional differences in the proteome of the early *Drosophila* embryo. We looked at a stage soon after the maternal-to-zygotic transition in gene expression, namely the onset of morphogenesis. We can compare our results to a previous study (*Gong et al., 2004*) on regional differences in the proteome at this stage that used mutants, as we did, to represent different regions of the embryo, and in that regard should be directly comparable. This study was based on 2D gel electrophoresis combined with mass spectrometry, which, while ground-breaking at the time, allowed only a small number (37) of unique proteins to be identified. All of these were also detected in our proteomes.

Because the differential detection in this study was based on PAGE, it was possible to detect different protein isoforms and therefore differences that may be due to phosphorylation. Of the proteins with variable isoforms, we found that 15 were phosphorylated in our own study, of which seven show differences in the mutants, and all of these are consistent with the changes seen in the 2D-PAGE experiment (*Gong et al., 2004*).

We also detected known phosphosites in proteins that act on the Rho-pathway, such as Sqh-T21, Sqh-S22 and Cofilin-S3 and differentially regulated phosphosites in proteins with key functions at the gastrulation stage, such as LIMK1 and RhoGEF2, and in setting up the DV axis, namely Toll and Cactus.

While it is reassuring to find phosphosites in known players in the early embryo, it is not clear whether those in Toll and Cactus, or their regional differences, allow us to infer new biological insights on the Toll signaling pathway from our current results. It is not clear what the spatial differences in the abundance of these phosphosites in Cactus and Toll signify, because the peak activity of the pathway is an hour before the timepoint we assay here. In the embryos we use, the transcriptional output, that is high expression of *twist* and snail, repression of zen etc., is fully established, and we may be seeing the effects of pathway down-regulation or feedback loops rather than signs of primary activity.

Comparing protein abundance against RNA expression could, in principle, reveal which proteins are post-transcriptionally regulated, but this can only be done if the techniques and approaches are as near-identical as possible, and if the results are technically perfect. Thus, even comparing differential RNA expression data obtained with different methods yields only partially overlapping results. For example, an Affymetrix-based study that again used mutants to represent regions along the DV axis of the embryo (*Stathopoulos et al., 2002*) identified 23 genes for which the RNA levels were higher in ventralized than in lateralized or dorsalized embryos. Comparing those to the expression patterns determined by single-cell RNA sequencing (*Karaiskos et al., 2017*) reveals that five appear to have little or no dorso-ventral modulation, a result that is also confirmed in the BDGP in situ hybridisation database. Those genes previously identified by genetic or functional studies, and known to be involved in mesoderm development (including marker genes like *twist*, *snail*, *zfh1*, *htl* etc) show up in all studies.

Thus, a comparison of our proteome data to reported RNA expression patterns has to be seen with caution. Nevertheless, such comparisons showed good matches for the abundant, well-studied genes and proteins: We detect the proteins for 13 of the 17 genes that are seen as ventrally upregulated genes in both studies (*Stathopoulos et al., 2002*; *Karaiskos et al., 2017*). Of those, we see all but four as ventrally upregulated, again including known ventral marker genes.

These comparisons lead to the question of how to judge which of the differences in protein abundance or regulation are biologically relevant and therefore interesting to follow up with functional studies. Confining the selection to those that are consistent with other studies would defeat the purpose of the experiment. Similarly, choosing the extent of change as a threshold would also exclude proteins we know to play a role in morphogenesis at this stage but which show only very small differences in expression. One example is Traf4 (*Mathew et al., 2011*), which is active in the mesoderm, but expressed there at low levels, and becomes expressed in the ectoderm as gastrulation begins. In our experiment, it was strongly downregulated in the dorsalized embryo, but showed only sub-threshold upregulation in the ventralized embryos.

To obtain a better picture of processes or cellular components involved in the functional differentiation of the cell populations, rather than looking at individual genes, we identified networks of functionally related proteins that were enriched among the differentially regulated entities. We would like to highlight here the mechanisms of differential protein degradation, mRNA regulation and microtubule modifications.

A role for protein degradation in creating differential functions along the DV axis has previously been illustrated by the case of the E3-ubiquitin ligase Neuralized (Neur) which is required and upregulated in the prospective mesoderm (*Perez-Mockus et al., 2017*). The network analysis identified the cullin complex as differentially expressed and differentially regulated (*Figure 6A*). We also find Neur in increased abundance ventrally. Known biological data thus validate the relevance of this network, which may in turn help to identify the as yet unknown targets for Neur in the mesoderm.

Another mechanism for post-transcriptional gene regulation is the differential translation or degradation of mRNAs along the dorso-ventral axis, and we find an enrichment of P-granule-related networks both in the proteome and the phosphoproteome. These networks are enriched within DV clusters with complete or partial ectodermal fate, that is the same clusters that show a strong uncoupling between mRNA and protein abundance (*Figure 6B*). Partial agreement between mRNA and protein spatial distribution is not an exclusive feature of the gastrula: it has also been described for larval tissues derived from the ectoderm and neuroectoderm, where nearly all studied genes show mRNA/protein discordance (*Titlow et al., 2023*; 97.5%; N=200 proteins). Therefore, the uncoupling between mRNA and protein abundance seems to be the rule rather than exception in at least these tissues, highlighting the importance of post-transcriptional regulation on gene expression regulation during development.

The diffused networks also showed phosphorylation of microtubules as a differentiating mechanism along the dorso-ventral axis during gastrulation, an interesting finding, because in epithelial tissues microtubules are often required for cell shape homeostasis (*Picone et al., 2010*; *Gomez et al., 2016*). Morphogenetic cell shape changes in *Drosophila* for which microtubules are essential include the squamous morphogenesis of the amnioserosa (*Pope and Harris, 2008*) and the invagination of the mesoderm (*Ko et al., 2019*) and the salivary placode (*Booth et al., 2014*). Here, we found that dorsal fold formation also requires microtubules. Ventral furrow and dorsal fold formation differ in their dependency on myosin (*Rauzi et al., 2015*; *Martin, 2020*), but our results show that both require microtubules for the basal relocalisation of nuclei. This requirement is functionally distinct from the association of microtubules with actomyosin during myosin-dependent tissue folding (*Ko et al., 2019*; *Booth et al., 2014*; *Manning et al., 2013*) and instead, may relate to the classic role of microtubules in vectorial trafficking and organelle localisation (*de Forges et al., 2012*; *Burute and Kapitein, 2019*). One reason why nuclei need to be actively repositioned may be that in their apical location they constitute a physical barrier to the cell's apical constriction.

## The differential proteomes and phosphoproteomes of the *Toll*[10B] and *spn27A* ventralizing mutants

Both *Toll*[10B] and *spn27A* ventralising mutations produced embryos that recapitulated known biological qualities of the mesoderm along the entire DV axis, such as the expression of ventral fate determinants,

or the apical localisation of the adherens junctions. However, these similarities were not fully mirrored in their proteomes. Curiously, the *spn27A* proteome seemed to be more similar to the dorsalized than to the *Toll^10B* proteome which would indicate that *Toll^10B* embryos are 'more' ventral than *spn27A* embryos. However, most of the mesodermal marker genes (*snail, twist, mdr49, wntD, neur*) make an exception are more abundant in *spn27A* embryos. Similarly ectodermal fate markers are more strongly downregulated in *spn27A* than in *Toll^10B* embryos. Specifically, *Toll^10B* mutants fail to repress the expression of ectodermal genes such as *egr, zen* and *crb*.

How can ventralizing mutations that act on the same pathway yield different proteomes? Spn27A is a serine protease inhibitor of the pathway that creates the active form of Spätzle (Spz), the ligand for Toll. Both mutations lead to constitutive activity of Toll, *Toll^10B* through a mutation in the receptor itself (*Schneider et al., 1991*; *Erdélyi and Szabad, 1989*), *spn27A* through enabling a homogeneously high level of Spz along the DV axis (rather than a peak on the ventral side; *Ligoxygakis et al., 2003*). Because Spz is highly abundant it should not be a limiting factor for the activation of Toll (*Morisalo and Anderson, 1995*; *Schneider et al., 1994*; *Figure 2—figure supplement 1A*) and loss of Spn27A should enable the full activation of Toll along the embryonic DV axis. Our results indicate that constitutively active Toll does not lead to the same level of signaling as the binding of the ligand to the receptor, and that these different levels lead to unexpected differences in the downstream targets of the signaling pathway.

## The biological significance of deviations from the linear model

The 'linear model' we formulated is based on the assumption that each mutant embryo faithfully represents one defined area of cells along the DV axis of the embryo, and that the full set of cell types in the embryo can therefore be reconstituted as the sum of the mutant cell types – weighted according to the area they occupy in the embryo. This should also be recapitulated for any individual protein expressed in the embryo. We found this to be true not only for the trivial cases of those proteins that occur at equal level in all genotypes, but also for most of the differentially modulated proteins. However, some proteins and phosphosites did not fit the model but showed strong deviations. One explanation could be that in the normal embryo the embryonic regions communicate with each other, and this communication is necessary for the expression or modification of certain proteins. These interactions cannot occur when the fates occur in isolation from each other in the mutants, and therefore some proteins would not be regulated properly and would not fit the model. Thus, wherever an interaction between the cell populations in the embryo is necessary for generating the correct expression or phosphorylation level, the linear model we proposed no longer applies; this means that strong deviations may indicate non-autonomous regulation. We do know some genes whose expression along the DV axis is determined by input from neighboring regions, such as Sog, Ind and single-minded. We indeed find that one of those proteins, Ind, is an outlier (deviation = 2.6) with higher than predicted expression in the dorsalized and ventralized mutants, consistent with repressive input from these regions in the wildtype.

Another case of proteins not following the linear model are those that are found in either decreased or increased abundance in all genotypes, a behavior we observed for a small percentage of proteins and phosphosites, perhaps as part of a general stress response related to the mutant situation. This is illustrated by the most extreme example, TM9SF4, which encodes an immune-related transporter that is present in all mutants at nearly 100-fold higher levels than in the wildtype. However, we did not find that this was a general rule either in the proteomes or in the phosphoproteomes: stress-related categories such as those from the 'chaperone' or 'immune response' ontology classes represented only a small percentage of proteins and phosphoproteins with the highest deviations (*Figure 3—figure supplement 1D, E*).

## Methods
### *Drosophila* genetics and embryo collections

*w^1118* (wildtype/control genotype in our studies, Bloomington stock 3605), *gd^9*/FM6a (*Konrad et al., 1988*) (provided by S. Roth), *Tl^rm9* and *Tl^rm10* (*Anderson et al., 1985*) (provided by A. Stathopoulos), *Toll^10B* (*Schneider et al., 1991*; *Erdélyi and Szabad, 1989*), *Df(3 R)ro80b/TM3* (Bloomington stock 2198), *spn27A^ex*/CyO (*Ligoxygakis et al., 2003*) (provided by S. Roth, Bloomington stock 6374),

*Df(2 L)BSC7/CyO* (**Ligoxygakis et al., 2003**) (provided by S. Roth). To visualize non-muscle myosin in vivo, a sqh-sqh::mCherry transgene (Bloomington stock 59024) was used to construct the following stocks *gd⁹;sqh-sqh::mCherry / CyO, sqh-sqh::mcherry /CyO;Tlʳᵐ⁹* and *sqh-sqh::mcherry / CyO;Df(3 R) ro80b/TM3*.

Dorsalized embryos were derived from *gd⁹* homozygous female mothers, lateralized embryos were derived from trans-heterozygous *Tlʳᵐ⁹/Tlʳᵐ¹⁰* mothers, ventralized embryos we derived from *Toll¹⁰ᴮ/Df(3 R)ro80b* and *spn27Aᵉˣ/Df(2 L)BSC7* mothers (see **Supplementary file 1**). Female mutant mothers were crossed with *w¹¹¹⁸* males, and the F1 from each of these crosses were collected and processed for mass-spectrometry analyses.

To visualize Myosin Light Chain, we generated the following mothers: Dorsalized: *gd⁹;sqh-sqh::mCherry/+,* Lateralized: *sqh-sqh::mcherry/+;Tlʳᵐ⁹/Tlʳᵐ¹⁰,* Ventralized: *sqh-sqh::mcherry/+;Toll¹⁰ᴮ/Df(3 R)ro80b.* Female mutant mothers were then crossed with *w¹¹¹⁸* males, and from F1 of these crosses, embryos in stage 5 a,b (**Campos-Ortega and Hartenstein, 1997**) were hand-selected under a dissecting microscope and mounted for live imaging (see below).

## Embryo collections

Embryos collected half an hour after egg-laying were allowed to develop for 2 hr 30' at 25 °C in a light and humidity-controlled incubator and then dechorionated in 50% bleach for 1' 30", washed with $H_2O$ and visually inspected under a dissecting microscope (Zeiss binocular) for 15'–20' at RT. To ensure younger embryos from each synchronized collection were in the target developmental stage (gastrulation stage, Stages 6 a,b **Campos-Ortega and Hartenstein, 1997**), we individually hand-selected the embryos on wet agar, which made the embryos semi-transparent, allowing the assessment of a range of morphological features, of which at least some are visible in each of the mutants:

- Yolk distance to embryonic surface: distinguishes between early (stage 5 a **Campos-Ortega and Hartenstein, 1997**) and late cellularization (stage 5b **Campos-Ortega and Hartenstein, 1997**).
- Yolk distribution within the embryo: identification of large embryonic movements of the germ band (eg.: Initiation of germ band extension, marking the initiation of stage 7 **Campos-Ortega and Hartenstein, 1997**). In DV patterning mutants, this is seen as twisting of the embryo.
- Change in the outline of the dorsal-posterior region: polar cell movement from the posterior most region of the embryo (stage 5 a/b **Campos-Ortega and Hartenstein, 1997**) to stage 6 a/b.
- Formation of the cephalic and dorsal folds: identification of stage 6 (**Campos-Ortega and Hartenstein, 1997**) (initiation of cephalic fold) and stage 7 (**Campos-Ortega and Hartenstein, 1997**) (dorsal folds).

The combined use of these morphological criteria, together with the synchronized egg collections allowed the accurate staging of wild type and mutant embryos. Any embryos that had developed beyond the initial stage of gastrulation (as judged by the abovementioned morphological criteria, Stage 7 **Campos-Ortega and Hartenstein, 1997**) were discarded and the remaining embryos were placed in 0.5 ml Eppendorf tubes and flash-frozen in liquid nitrogen. An 0.5 ml Eppendorf tube filled with embryos yields approximately 1 mg of protein.

## Transgenic fly lines generated in this work

Three transgenic lines were generated to visualize cell membranes and microtubules. For cell membranes, a single copy of EGFP or three copies of mScarlet interleaved with linkers (DELYKGG-GGSGG) were trailed by a C-terminal CaaX sequence from human KRas4B (KKKKKKSKTKCVIM) for membrane targeting to yield EGFP-CaaX or 3xmScarlet-CaaX. For microtubules, EMTB-3xGFP from addgene #26741 was cloned into pBabr, a ΨC31 site-directed transformation vector, between the maternal tubulin promoter and the spaghetti-squash 3' UTR (**Takeda et al., 2018**). These constructs were then integrated into the fly genome at attP2 or attP40 by Rainbow Transgenics Flies, USA, or WellGenetics, Taiwan.

## SILAC metabolic labeling

SILAC metabolic labeling was performed using yeast transformed to produce heavy lysine (SILAC yeast with LysC13/6, Silantes: https://www.silantes.com/). To standardize each of the phosphoproteomic runs for each condition we labeled the proteome of *w¹¹¹⁸* control flies (**Figure 1—figure supplement**

*1D*). Therefore, our analyses included a SILAC-labeled and an unlabelled $w^{1118}$ extract. To maximize the incorporation of LysC13/6 into the *Drosophila* proteome, unlabelled $w^{1118}$ adult flies were raised in bottles prepared with SILAC yeast as the only source for amino acids (2% SILAC yeast). The fly media were prepared in agreement with the recommendations of Silantes (https://www.silantes.com/). All emerging larvae were fed on SILAC fly medium from L1 until adult stage, in a temperature (25 °C), light and humidity-controlled incubator (Sanyo). The emerging labeled $w^{1118}$ adults were then transferred to cages for embryo collections, and fed with wet SILAC yeast until disposal of flies (after 2–3 weeks). SILAC $w^{1118}$ embryos were collected as described above.

Using this protocol, we labeled ~75% of the proteome of SILAC $w^{1118}$ embryos. SILAC labeling did not affect the phosphoproteome of wild type embryos, and had only a minor effect on phosphosite intensity distribution, indicating standardization with Lys 13/6 was a valid approach (*Figure 1—figure supplement 1E*).

## Proteomic analyses

### Protein digestion

Embryos were lysed in 6 M urea and 2 M thio-urea (100 mM HEPES pH = 8.5). Lysates were treated by ultrasonic (20 s, 1 s pulse, 100% power) on ice and cleared by centrifugation (15 min, 22 °C, 12,500 x *g*). Protein concentration was determined with a DC Protein Assay (Bio-Rad).

For each proteome analyzes, a 200 µg sample was utilized. We analyzed for each genotype at least 3 technical replicates. Embryos derived from *spn27a*$^{ex}$/*df* and *gd*$^9$ mutant mothers were analyzed using two biological replicates with three technical replicates each, making a total of six analyzed replicates for these two genotypes. Proteins were reduced by Dithiothreitol (22 °C, 40 min) followed by protein alkylation using iodoacetamide (22 °C, 40 min in the dark). Lys-C endopeptidase was added for 2 hr at 22 °C. The samples were then diluted to 2 M urea using 50 mM ammonium bicarbonate. Trypsin was added in a 1–100 enzyme:substrate ratio and incubated overnight at 20 °C. Digestion was stopped by acidification using TFA at a final concentration of 0.5%. The resulting peptides were desalted using Waters SPE Columns (C18 material, 50 mg). Peptides were eluted with 60% acetonitrile and 0.1% formic acid. The eluate was dried using a SpeedVac concentrator (Eppendorf) to complete dryness. Peptides were then separated by offline high-pH fractionation.

For phosphopeptide enrichment a SILAC-based quantification was applied. For each sample, 500 µg of protein lysate was mixed with an equal amount of Lys-6 SILAC labeled protein lysate and digested as described above except that Lys-C instead of trypsin was used exclusively. We split the protein lysate from each population of embryos in three and conducted three separate analyses (digest, PTM enrichment, LC-MS/MS). The peptide solution was desalted using Waters SEP-PAK 50 mg C18 cartridges and then subjected for phosphopeptide enrichment.

### High-pH HPLC offline fractionation

The instrumentation consisted of an Agilent Technologies 1260 Infinity II system including pumps (G7112B), UV detector (G7114A), and a fraction collector (G1364F). A binary buffer system consisting of buffer A, 10mM ammonium hydroxide in 10% methanol and buffer B (10mM Ammonium hydroxide in 90% acetonitrile) was utilized. Peptides (resuspended in buffer A) were separated on a KINETEX EVO C18 2x150mm column using a flow rate of 250 µL/min and a total gradient time of 65 min. The content of buffer B was linearly raised from 2% to 25% within 55 min followed by a washing step at 85% buffer B for 5 min. Fractions were collected every 60 s in a 96-well plate over 60 min gradient time collecting a total number of 8 fractions per sample. Before each run, the system was equilibrated to 100% buffer A. The fractions were then concentrated in a SpeedVac concentrator (Eppendorf) and subjected to an additional desalting step using the StageTip technique (SDB-RP, Affinisep). Prior to LC-MS/MS measurement, peptides were solubilized in 10 µL of 2% formic acid and 2% acetonitrile. 3 µL were injected per LC-MS/MS run.

## Phosphopeptide enrichment

For phosphopeptide enrichment, the High-Select TiO2 Phosphopeptide Enrichment Kit (#A32993) was utilized following the manufacturer's instructions. In brief, desalted peptides were dried to complete dryness and resuspended in Binding Buffer (included in kit). The peptide solution was centrifuged (10 min, 12.500 x *g*, 22 °C) and the supernatant was transferred to TiO2 tips. Phosphopeptides

were enriched and eluted using the provided elution buffer. The eluate was immediately dried in a SpeedVac concentrator (Eppendorf) and stored at –20 °C. Prior to LC-MS/MS measurement, peptides were solubilized in 10 µL of 2% formic acid and 2% acetonitrile. Three µL were injected per LC-MS/MS run. The phosphopeptide enrichment was performed in technical duplicates.

## Liquid chromatography and mass spectrometry

The LC-MS/MS instrumentation consisted of a nano-LC 1000 coupled to a QExactive Plus or of a nano-LC 1200 (Thermo Fisher) coupled to a QExactive HF-x instrument via electrospray ionization. The buffer system consisted of 0.1% formic acid in water (buffer A) and 0.1% formic acid in 80% acetonitrile. The column (75 µm inner diameter, 360 µm outer diameter) was packed with PoroShell C18 2.7 µm diameter beads. The column temperature was controlled to 50 °C using a custom-built oven. Throughout all measurements, MS1 spectra were acquired at a resolution of 60,000 at 200 m/z and a maximum injection time of 20ms was allowed. For whole proteome measurements, the mass spectrometer operated in a data-dependent acquisition mode using the Top10 (QExactive Plus) or Top22 (QExactive HF-x) most intense peaks. The MS/MS resolution was set to 17,500 (QE-Plus) or 15,000 (QE-HFx) and the maximum injection time was set to 60ms or 22ms, respectively. Samples of replicate one and three were measured on the QE-Plus system and replicate two was measured on the QE-HF-x system.

For phosphoproteome analysis, the MS2 resolutions were set to 30,000 (QEx-Plus) or 45,000 (QEx-HFx). Samples of all three replicates were measured on the QEx-HFx system. We added trial samples measured on the QEx-Plus system to increase the phosphosite coverage.

## Proteomic and phosphoproteomic data analysis

Raw files were processed using MaxQuant (v. 1.5.3.8; *Cox and Mann, 2008*) and the implemented Andromeda search engine (*Cox et al., 2011*). The Uniprot reference proteome for *Drosophila melanogaster* (downloaded: 07.2016, 44761 entries) was utilized. Phosphoproteome and proteome data were analyzed separately and the match-between-runs algorithm was enabled. For proteome analysis, the label-free quantification method (MaxLFQ) was enabled using the default settings. Default settings for the mass tolerances for FTMS analyser were used. The FDR was controlled using the implemented (Andromeda) reverse-decoy algorithm at the protein, peptide-spectrum-match and PTM site levels to 0.01.

For SILAC-based phosphopeptide quantification, a minimum ratio count of 2 was required; the minimum score for the modified peptide was set to 20. Technical duplicates were aggregated by using the log2 normalized SILAC ratio median.

The proteinGroups (proteome) and PhosphoSite(STY) tables were subjected to downstream analysis. Gene Ontology annotations were derived from the Uniprot database and annotated. The LFQ intensities were log2 transformed. Pairwise comparisons were performed using a two-sided unpaired t-test. One-way Analysis of Variance (ANOVA) was performed on genotypes and a FDR was calculated using a permutation based approach (s0=0.1, #permutations = 500) in the Perseus software (*Tyanova et al., 2016*).

## Matching and correlation between proteome and phosphoproteome

Protein log2 fold change ratios were matched to the phosphosite table using the Uniprot identifiers. If the phosphorylation site was part of multiple protein groups, the average log2 fold change was utilized. The analysis of the correlation between the fold changes of phosphosites and their host proteins was performed as follows: for each proteome-matched phosphosite, a protein-phosphosite pair was assembled, yielding 6297 pairs of phosphosites and their host proteins taking into account all genotypes. We tested for each differentially phosphorylated site whether its respective protein was up- or down-regulated and made this comparison for each mutant genotype versus the wildtype. To consider a protein and a phosphosite regulated, we applied the same thresholds as used for clustering: ± 1.4 (0.5 in log2) fold change for proteins, and ± 1.3 (0.35 in log2) fold change for phosphosites (see below Hierarchical clustering analyses, Threshold determination). The protein-phosphosite pairs were placed in a scatter plot with 4 quadrants that were connected to 3 possible behaviors: correlation (fold change of host protein and phosphosite are both positive -green- or negative -blue-), anti-correlation (fold change of host protein and phosphosite have different signs -red/magenta-) or

no-correlation, (fold change of host protein is within threshold range but phosphosite trespasses it -black/white-). Finally, we counted the number of protein-phosphosite pairs that displayed each of these described behaviors, and this quantification was used to assemble the bar plots.

## Immunostainings and live imaging procedures

### Synchronized egg collections
Eggs were collected for 1 hr, allowed to develop for a further 2 hr 30' in a temperature (25 °C), light and humidity-controlled incubator (Sanyo) and then dechorionated in sodium hypochlorite (50% standard bleach in water) and washed thoroughly with water. Depending on the type of staining and antigen, embryos were fixed using the appropriate standard protocols.

### In situ RNA hybridisation
Antisense probes for Dpp, Sog, and Snail were used on dechorionated embryos by applying *Drosophila* standard protocols for in situ hybridisation with digoxigenin-labeled RNA-probes (*Tautz and Pfeifle, 1989*).

### Heat fixation for imaging of Armadillo/β-catenin
Dechorionated embryos were transferred to a beaker containing 10 ml of boiling heat-fixation buffer (For 1 L in water: 10 X Triton-Salt Solution, 40 g NaCl, 3 ml Triton X-100 (T9284 Sigma)), and fixed for 10 s. Fixation was stopped by placing the beaker containing the embryos on ice. Vitelline membranes were removed by transferring the embryos to a tube containing heptane:methanol (1:1), vortexed for 30 s and rehydrated.

### Fixation for imaging of microtubules
To visualize microtubules, a formaldehyde-methanol sequential fixation was performed as previously described (*Gomez et al., 2016*). Dechorionated embryos were fixed in 10% formaldehyde (methanol free, 18814 Polysciences Inc) in PBS:Heptane (1:1) for 20 min at room temperature (RT), and devitellinised for 45 s in 1:1 ice-cold methanol:heptane. Embryos were stored for 24 hr at –20 °C and rehydrated before use.

## Antibody staining procedures
Rehydrated embryos were blocked for 2 hr in 2% BSA (B9000, NEB) in PBS with 0.3% Triton X-100 (T9284 Sigma). Primary antibody incubations were done overnight at 4 °C. Primary antibodies used were: mouse anti α-tubulin 1:1000 (T6199, clone 6-11B-1, Sigma), mouse anti acetylated-α-tubulin FITC conjugated 1:250 (sc23950, Santa Cruz Biotechnology), rat anti tyrosinated-α-tubulin 1:250 (MAB1864-I, clone YL1/2, EMD Millipore/Merck), mouse anti-Armadillo/β-Catenin 1:50 (N27A1, Developmental Studies Hybridoma Bank), rabbit anti Snail (*Tyanova et al., 2016*) 1:500. Incubations with secondary antibodies were performed for 2 hr at RT. Alexa Fluor 488- and 594-coupled secondary antibodies were used at 1:600 (488 and 594 Abcam).

## Preparation of physical cross-sections
Immunostained embryos embedded in Fluoromount G (SouthernBiotech 0100–01) were visually inspected under a dissecting microscope (Zeiss binocular) to select the desired developmental stages. The embryos were sectioned manually with a 27 G injection needle at approximately 50% embryo length and slices were mounted for microscopy.

## Image acquisition
Images in *Figure 1C* were acquired with a Zeiss LSM880 Airyscan microscope, using a Plan-Apochromat 63 x oil (NA 1.4 DIC M27) objective at 22 °C, with a z-slice size = 0.3 μm. Acquired volumes were max-projected (along z axis) for a range of 1.5 μm (5 slices). Images in *Figure 7A* and *Figure 7—figure supplement 1B* were acquired using a super resolution Deltavision OMX 3D-SIM (3D-SIM) V3 BLAZE from Applied Precision (a GE Healthcare company). Deltavision OMX 3D-SIM System V3 BLAZE is equipped with 3 sCMOS cameras, 405, 488, and 592.5 nm diode laser illumination, an Olympus Plan Apo 60X1.42 numerical aperture (NA) oil objective, and standard excitation and emission filter sets.

Imaging of each channel was done sequentially using three angles and five phase shifts of the illumination pattern. The refractive index of the immersion oil (Cargille) was 1.516. Acquired volumes were max-projected (along the z axis) for a range of 3 μm (10 slices). Images in *Figure 7B* and *Figure 7—figure supplement 1A* were acquired with a Leica SP8 microscope equipped with white laser. Gated detection on HyD detectors was used for each shown channel using a Plan-Apochromat 63 x oil (NA 1.4) objective at 22 °C, with a z-slice size = 0.3 μm. Acquired volumes were max-projected (along z axis) for a range of 1.5 μm (5 slices).

## Live imaging of myosin light chain

Dechorionated embryos expressing maternally provided sqh::mCherry were mounted in 35 mm glass-bottom petri dishes in two different ways with either the embryonic dorsal, lateral, or ventral surface facing the glass bottom or vertically glued (heptane glue) on their posterior end to the glass-bottom and embedded in 0.8% low melting point agarose in PBS that was previously cooled to 30 °C. For the superficial imaging of the sub-apical domain of embryos along their dorsal, lateral and ventral sides, we acquired volumes of 20 μm (z-slice size of 0.8 μm) using a PerkinElmer Ultraview ERS (microscope stand Zeiss Axiovert 200) with a Yokogawa CSU X1 spinning disk with a Plan-Apochromate 63 x (NA 1.4, oil) objective at 22 °C. The vertical mounting enabled the imaging of myosin light chain along the dorso-ventral cross-section of living embryos (*Krueger et al., 2020*) in the x-y plane. We acquired 1–3 slices (z-slice size = 1,1 μm) at 130–150 μm from the posterior end of the embryo using a Zeiss LSM780 NLO 2-photon microscope with a Plan-Apochromat 63 x objective (NA 1.4, oil, DIC M27) at 22 °C.

## Calculation of correlation matrices

Correlation matrices were calculated using Matlab R2019b on the proteins and phosphosites that were detected in all genotypes and all the replicates. For this, we used the 'corr' function to calculate the Pearson correlation between log2 intensities of proteins or phosphosites in the replicates of all genotypes. The resulting correlation matrix was plotted using the 'clustergram' function by applying the following settings: linkage: average; RowPDist & ColumnPDist: correlation.

## Development of a linear model for protein and phosphosite abundances

We defined a 'linear model' that is based on the assumption that the three types of mutant embryos (dorsalized, lateralized, and ventralized) each represent one region along the DV axis of the embryo, and that protein abundance in the three regions should add up to the total protein abundance in the entire embryo, and therefore, the abundance in the three mutants should add up to the same value, if each is weighted by the region occupied in the wildtype embryo.

This linear model can be expressed for each protein ProtX as a sum, $^{t}wt_{ProtX}$, where D, L, and V are the abundance of a protein 'ProtX' in the proteomes of the three mutant genotypes (means of the log2 intensity values of the replicates, transformed to its linear value), and a, b, and c represent the proportion of each region along the dorsoventral axis:.

$$^{t}wt_{ProtX} = a^{*}D + b^{*}L + c^{*}V$$

This model requires values a, b, and c for the three regions along the DV axis. For the mesoderm, this has been reported as 0.2 from measurements on cross-sections (*Rahimi et al., 2016*), but we wanted to determine the theoretical optimum for each of the values without any prior assumption about their real proportions in the embryo. The theoretical optimum would be one for which the proportions for the three regions when used in the sum yield a theoretical ('t') value $^{t}wt_{ProtX}$ that is the closest to the experimentally measured ('m') value $^{m}wt_{ProtX}$ for that protein in the wildtype embryo.

To systematically explore the proportions for each region, we tested all possible combinations for a, b, and c at 0.05 steps in the range from 0 to 1 (i.e.: 0, 0.05, (...), 0.95,1). For each calculated $^{t}wt_{ProtX}$, we calculated the deviation from the measured abundance by subtracting the mean linear intensity measured for the same protein in a wild type embryo ($^{m}wt_{ProtX}$), and transforming this difference between the theoretical and measured wild type to log2 scale as follows:

$$Deviation_{ProtX} = log2(^{t}wt_{ProtX}/^{m}wt_{ProtX})$$

A log2 value of 0 for the deviation therefore indicates a perfect match between the theoretical calculation and the actual measurement. For each possible combination of a, b, and c we obtained distributions of Deviation$_{ProtX}$ values, for which we calculated the Interquartile Range (IQR) using the IQR function from Matlab 2019b. Based on the assumption that the best matching proportions should lead to the narrowest dispersion of the distribution of Deviation$_{ProtX}$, we sorted (smaller to largest) the combinations of proportion constants based on their calculated IQRs. This parameter screen yielded good fits for a range of combinations. Previous work indicated the mesoderm represents 20% of the circumference of the embryo *Rahimi et al., 2016*; however, for the two best, the area of the ventral region was slightly larger than the observed 20% of the circumference of the embryo in vivo (*Figure 3—figure supplement 1B, C*, *Supplementary file 5*). The third best was one for which the ventral domain corresponded to the experimentally measured value of 0.2 (20%, *Figure 3—figure supplement 1B, C*, *Supplementary file 5*), and for the lateral and dorsal domains the value was 0.4, which matched estimations based on the expression domains of lateral and dorsal genes (*Stathopoulos et al., 2002*; *Jaźwińska et al., 1999*). We therefore chose this set:

$$^{t}wt_{ProtX} = \mathbf{0.4}^{*}D + \mathbf{0.4}^{*}L + \mathbf{0.2}^{*}V$$

Because we used two different ventralising genotypes (*Toll$^{10B}$* and *spn27A$^{ex}$*), the $^{t}wt_{ProtX}$ was calculated twice for each protein, one for for each mutant genotype combination (i.e.: D-L-V$_{Toll10B}$ and D-L-V$_{spn27Aex}$).

## Hierarchical clustering analyses

### Data generation and hierarchical clustering

We included in this analysis all proteins that were detectable in the wildtype (5883/6111), even if they were undetectable in one or more mutant populations. To obtain clusters that represented the behaviors of proteins and phosphosites with respect to the wild type genotype, we took for each protein and phosphosite the mean log2 fold change. For this, we first calculated the mean log2 intensity values per protein and genotype, and next, calculated the log2 fold changes (FCs) for each protein and phosphosite between each DV mutant genotype and the wild type. Because we wanted to cluster exclusively by the direction but not the extent of changes between the DV mutant genotypes and the wild type, we assigned to each FC a value (1,–1 or 0) based on exceeding a FC threshold (see below '*Threshold determination*'). For proteins, the threshold was +/-1.4 (0.5 in log2) FC, for phosphosites was +/-1.27 (0.35 in log2) FC. When proteins and phosphosites exceeded the FC threshold, we assigned a+1 or –1 for positive and negative FCs respectively. When proteins and phosphosites remained within the threshold range (i.e.: for proteins: –1.4 (0.5)< FC < 1.4 (0.5); for phosphosites: –1.27 (0.35)<FC < 1.27 (0.35)), we assigned a 0. For proteins and phosphosites that were undetected in a particular DV patterning mutant, we assumed -based on the detection of Twist and Snail across mutant genotypes-, these were in decreased abundance vs. the wild type, and we assigned to these –1.

Next, we filtered the set of proteins and phosphosites that we used as a source for the hierarchical clustering. Proteins and phosphosites with a 0 assigned in all FC comparisons (i.e. D vs. WT = 0; L vs. WT = 0, Vtl vs WT = 0, Vsp vs WT = 0) were considered unchanged in our study and therefore were filtered out (number proteins = 2156/6111; number phosphosites = 1234/6259). Proteins and phosphosites with a+1 or a –1 assigned to all FC comparisons (i.e.: D vs. WT = 1 (-1); L vs. WT = 1 (-1), Vtl vs WT = 1 (-1), Vsp vs WT = 1 (-1)) were also filtered out (number proteins = 329/6111; number of phosphosites = 615/6259). We proceeded with the clustering of the rest of the proteins (3398/6111) and phosphosites (3433/6259).

The thresholded FCs of the filtered set of proteins and phosphosites were transformed in row z-scores (i.e.: calculated per protein and per phosphosite). The reason for this is that this method takes into account that value sets that represent similar relative differences between the mutants (for example, 0 –1–1 vs. 1 –1–1 or 1 0 0) are biologically more similar to each other than the raw values indicate. The z-scores for all of these cases would be 1.1547 –0.5774–0.5774. The hierarchical clustering was conducted both on rows (proteins or phosphosites) and columns (FC of each mutant genotype vs. the wildtype) using the 'clustergram' function (Matlab R2019b) setting the linkage parameter in 'average' and the row probability distance parameter in 'Euclidean'. The output of the clustering in

the figures was set to be displayed using the mean log2 FCs of proteins and phosphosites between the DV mutant genotypes and the wild type.

## Threshold determination

We determined the threshold to be applied to the FCs between the DV mutant genotypes and the wild type of each proteomic experiment by analyzing the variability of the FCs within each biological replicate. We first took the mean log2 intensity per protein for the wildtype '$^{WT}Mean_{ProtX}$', and next, calculated the log2 FC between the log2 intensity of each technical replicate with measurable intensity for each biological replicate and the $^{WT}Mean_{ProtX}$ as follows:

$$^{BP-G}FC(ProtX)_{TR-N} = {}^{BP-G}log2(IntensityProtX)_{TR-N} - {}^{WT}Mean_{ProtX}$$

where BP-G is the biological replicate of a particular genotype and TR-N a technical replicate (1<N < 3) for a biological replicate BP-G. Therefore, we obtained a set of $^{BP-G}FC(ProtX)_{TR-N}$ values per protein, and the number of $^{BP-G}FC(ProtX)_{TR-N}$ values per protein equals the number of technical replicates with measurable intensity for a particular biological replicate.

Next, we calculated $^{BP-G}stdev(ProtX)$, which is the standard deviation of the set of $^{BP-G}FC(ProtX_{TR-N}$ values per protein and for each biological replicate. In this way, we obtained a distribution of the standard deviation of FCs per biological replicate. For each biological replicate distribution of FCs standard deviations we calculated the IQR (Prism Graphpad V8) and extracted the 3rd quartile value ($^{BR-G}Q3$) to capture up to 75% of the variability of each distribution:

| Biological Replicate Proteome | 3rd Quartile Value (Q3) |
| --- | --- |
| **Dorsalized (replicates 1–3)** | **0.543** |
| **Dorsalized (replicates 4–6)** | **0.492** |
| Lateralized | 0.471 |
| Ventralized *spn27A* (replicates 1–3) | 0.463 |
| Ventralized *spn27A* (replicates 1–6) | 0.481 |
| Ventralized *Toll^{10B}* | 0.565 |

| Biological Replicate Phosphoproteome | 3rd Quartile Value (Q3) |
| --- | --- |
| Dorsalized | 0.405 |
| Lateralized | 0.366 |
| Ventralized *spn27A* | 0.377 |
| Ventralized *Toll^{10B}* | 0.259 |

Finally, we defined the FC threshold as the mean of the Q3 value across the biological replicates of each experiment (number of: biological replicates proteome = 6; biological replicates phosphoproteome = 4) as follows:

$$^{Proteome}FC\_Threshold = sum(^{BR-G}Q3)/6 = 0.503$$

$$^{Phosphoproteome}FC\_Threshold = sum(^{BR-G}Q3)/4 = 0.352$$

## RNA-proteome comparison along DV cell populations

### Data generation: NovoSpark analyses of single-cell RNAseq data

The single-cell RNAseq data derived from stage 6 *Drosophila* embryos (*Karaiskos et al., 2017*), were spatially reconstructed with novoSpaRc (*Latapy, 2016*). As prior spatial information 84 known gene expression patterns were used from the BDTNP atlas (downloaded from: https://shiny.mdc-berlin. de/DVEX/). NovoSpaRc embeds each cell probabilistically over 3039 locations using a generalized optimal-transport approach. This results in a 'RNA Atlas', which includes a predicted spatial gene expression pattern for every detected gene.

## RNA clustering

We excluded all genes that were scored as ubiquitously expressed in THE RNA ATLAS (*Karaiskos et al., 2017*). Of the remaining 8924 genes, we selected those that were also listed in our clustered proteomic dataset (3346 genes coding for 3383 proteins), yielding a list of 3086 non-ubiquitous genes that were present both in the RNA atlas and the clustered proteome. These are sorted into classes by comparing the expression pattern of each to that of six reference genes with restricted dorso-ventral expression that represent the six regulation categories D, L, V, DL, DV, and VL. We used as reference genes *dpp* for dorsal, the average between *sog* and *soxN* for lateral, *twist* for ventral, *crb* for dorsal +lateral, *net* for dorsal +ventral and *neur* for lateral +ventral (*Figure 4A*). To compare similarity for each of the 3086 genes we calculated their spatial Pearson correlation to each of the reference genes. Each gene was then classified as belonging to the category of the reference gene for which the Pearson correlation was the highest. We therefore obtained 6 clusters of genes, which we termed 'DV RNA clusters' each of them with their corresponding maximum Pearson spatial correlation value (*Figure 4—figure supplement 1*). To filter out false positives, we selected those genes with the largest similarity to the reference genes, for which we expected strong differential expression along the dorso-ventral axis. We did this by determining the Pearson spatial correlation value corresponding to the 95th Percentile of each dorso-ventral RNA cluster, using the 'prctile' function (Matlab R2019b). Finally, we used the 95th percentile value as a threshold to filter the 5th percentile of genes from each DV RNA cluster. We obtained a list of 155 genes that we used to compare against the proteome clusters (*Supplementary file 9*), which we termed 'DV RNA Reference Sets'.

## RNA-proteome comparison

The filtered set of 155 genes codes for 157 proteins. We grouped the 157 proteins based on their DV cluster assignment and for each DV cluster, we classified its proteins based on the RNA reference set to which their genes had been allocated. Theoretically, if both classifications, i.e. the RNA reference set and the proteomes, were perfectly correct, then genes from a protein cluster should be included only in the corresponding RNA reference set. For DV clusters 1–12, we classified the results of this comparison as 'perfect match if the RNA expression pattern and the DV cluster belong to the same regulation category, as 'partial match if the RNA expression pattern and the DV cluster coincided only in one DV domain, or as mismatch if the RNA expression pattern and the DV cluster belong to mutually exclusive regulation categories.

## Calculation of euclidean distance score

We developed a score based on a calculated Euclidean distance to measure the proximity of each protein in a particular DV cluster to the most extreme fold changes measured in DV mutant vs. wild type comparisons. We used the same approach for the phosphosites.

We first calculated the mean log2 fold change (FC) between the DV mutants and the wild type (which meant we could not assess proteins nor phosphosites that were not detected in the wild type). Next, we rescaled each of the FC distributions [dorsalized (D) vs. WT, lateralized (L) vs. WT, ventralized $Toll^{10B}$/def (Vtl) vs. WT and ventralized $spn27A^{ex}$/def (Vsp) vs. WT] to transform the log2 FC = 0 in the original distributions as log2FC = 0.5 in the new, rescaled distribution. We first identified the upper and lower limits of each FC distribution, and next, transformed each pair (upper and lower) of limits to their absolute values. This enabled the identification of the largest absolute limit for each FC distribution, and depending whether the upper or the lower limit was the absolute largest, we used one of the following equations to rescale:

If the upper limit of a particular FC distribution is the absolute largest:

i. log2FC_rescaled_i = (log2FC_original_i + |max_log2FC_original|) / (2* |max_log2FC_original|)

If the lower limit of a particular FC distribution is the absolute largest:

ii. log2FC_rescaled_i = (log2FC_original_i + |min_log2FC_original|) / (2* |min_log2FC_original|)

where 'i' is a particular protein, and max_log2FC_original and min_log2FC_original are the upper and lower limit values for a particular FC distribution. Using this rescaling approach, we obtained a new set of rescaled FCs for each distribution (D vs. WT, L vs. WT, Vtl vs. WT and Vsp vs. WT). We

considered those proteins that were undetected in a mutant genotype as being in decreased abundance in that genotype, and imputed the lower limit value of the rescaled FC distribution for that genotype.

For each protein, we assigned two vectors with their corresponding rescaled FCs (shown here for one ventralized genotype, but also calculate separately for the other):

$$(D\_rescaledFC\_i, L\_rescaledFC\_i, Vtl\_rescaledFC\_i)$$

Next, we assembled reference vectors representing the most extreme behaviors for each regulation category (*Figure 3D*, upper panel), using the rescaled FCs:

| Regulation category | DV cluster | Reference vector components | Reference vector values with Vtl* | Reference vector values with Vsp* |
|---|---|---|---|---|
| Dorsal | 1 | (max$^{rFC}$,min$^{rFC}$,min$^{rFC}$) | (1, 0, 0.0047) | (1, 0, 0.0485) |
| Lateral | 2 | (min$^{rFC}$,max$^{rFC}$,min$^{rFC}$) | (0.1127, 0.9865, 0.0047) | (0.1127, 0.9865, 0.0485) |
| Ventral | 3, 4, 5 | (min$^{rFC}$,min$^{rFC}$,max$^{rFC}$) | (0.1127, 0, 1) | (0.1127, 0, 1) |
| Dorsal +Lateral | 6 | (max$^{rFC}$,max$^{rFC}$,min$^{rFC}$) | (1, 0.9865, 0.0047) | (1, 0.9865, 0.0485) |
| Dorsal +Ventral | 7, 8, 9 | (max$^{rFC}$,min$^{rFC}$,max$^{rFC}$) | (1, 0, 1) | (1, 0, 1) |
| Lateral +Ventral | 10, 11, 12 | (min$^{rFC}$,max$^{rFC}$,max$^{rFC}$) | (0.1127, 0.9865, 1) | (0.1127, 0.9865, 1) |

Where max$^{rFC}$ and min$^{rFC}$ are the maximum and minimum rescaled FCs for the corresponding distributions (D vs. WT, L vs. WT, Vtl/Vsp vs. WT). Finally, we calculated the Euclidean distance score (ED Score) between each protein in clusters 1–12, and the reference vectors that correspond to each DV class:

$$EDScore = sqrt[(D\_rescaledFC\_i - D\_ref)^2 + (L\_rescaledFC\_i - L\_ref)^2 + (V\_rescaledFC\_i - VL\_ref)^2]$$

Where D_ref, L_ref and V_ref are the reference vector values* for the corresponding regulation category.

## Ontology analyses using diffused networks
### Data generation
The method employed here is similar to the one developed in *Giudice et al., 2024*. Briefly, we retrieved the *Drosophila* protein-protein interaction network from IntAct (last update June 2020). We modeled the edge weights (*Resnik, 1999*) using the Resnik semantic similarity, which was calculated using the Semantic Measures Library (*Harispe et al., 2014*). We also generated 1000 random networks, where the node degrees are conserved, employing the vl method from the igraph library (*Latapy, 2016*). The edge weights of the random network are updated accordingly. To correct for the hub bias, we applied the Laplacian normalization to all networks using:

$$w_{ij} = \frac{w_{ij}}{\sqrt{d_i d_j}} \tag{1}$$

where $w_{ij}$ indicates the edge weight and $d_i$ and $d_j$ represent the weighted degree of node *i* and node *j* respectively. We extracted from the Pfam (*Mistry et al., 2021*) database (last update June 2021), all the kinases detected in *Drosophila* by selecting the CL0016 clan. Next, we employed the UniprotKB database to distinguish the tyrosine kinases (family: PF07714) from other kinases. In total 56 tyrosine kinases and 251 other kinases are present in the network. We also precalculated the mean and the standard deviation of the Resnik semantic similarity of each regulated node in the network against each other.

## Seed selection and network diffusion
We applied the random-walk-with-restart-based algorithm (*Giudice et al., 2024*) for the following DV clusters: D (1), L (2), V (5), DL (6), DV (9) and LV (12), once each for the ED score and once for the deviation values, and each for the protein and phosphoproteomic datasets. In the case of the deviation

values, we used only those proteins or phosphosites within the interquartile range. We assigned as seed value the reciprocal of the absolute value from both the ED Score or the log2 Fold Change Deviations. Note that if multiple phosphosites are assigned to the same protein we selected the median of the ED scores or log2 Deviations. We then partition the seed set in tyrosine kinases, remaining kinases and other proteins and perform the random walk with restart (RWR) from each of the three partitions separately. We also repeated the same procedure with the same set of initial nodes against 1000 random networks. We estimated the empirical p-value for each node of the network as the percent of its random scores that exceed the real score and selected only the nodes with a p-value <0.05 in at least one of the three partitions. The resulting subnetworks are further filtered using the ego decomposition (*Giudice et al., 2024*). Briefly, for each seed node we extracted ego networks with a maximum distance of 2 steps from the ego. We then filtered the ego networks by selecting only the most similar functional nodes to the ego using this formula:

$$z-score = \frac{Resnik\,(ego,\,j) - mean_{Resnik\,ego}}{std_{Resnik\,ego}} \tag{2}$$

where $Resnik\,(ego,j)$ is the semantic similarity between the ego and a node j in the ego network, $mean_{Resnik\,ego}$ and $std_{Resnik\,ego}$ are the mean and the standard deviation of the ego against all the other nodes in the initial network. Nodes with a z-score>1.28 (equivalent to a p-value <0.1) are retained. After this filtering, the resulting ego networks with less than 5 nodes are discarded. Additionally, the weight of the ego networks are changed according to (3) to reflect the functional impact of the dysregulation of the ego on the neighboring nodes.

$$w(i,\,j) = \begin{cases} Resnik\,(ego,\,j)\,if\,i = ego\,and\,j = \Gamma_{ego}\,or\,i = \Gamma_{ego}\,and\,j = \Gamma_{\Gamma_{ego}} \\ \dfrac{Resnik\,(ego,\,i) + Resnik\,(ego,j)}{2}\,if\,i = \Gamma_{ego}\,and\,j = \Gamma_{ego}\,or\,i = \Gamma_{\Gamma_{ego}}\,and\,j = \Gamma_{\Gamma_{ego}} \end{cases} \tag{3}$$

where $\Gamma\,(ego)$ represents the nodes at distance 1 from the ego and $\Gamma_{\Gamma(ego)}$ represents the nodes at distance 2 from the ego. The ego networks obtained are normalized again to correct for hubs using the Laplacian normalization using (1). For each ego network, we then calculate the topological distance vector and the functional distance vector as in Giudice et al. The topological distance vector is calculated using the following formula (4):

$$topological\,distance = 1000 \times log_2\left(1 - jsd\left(RWR_{node},RWR_{ego}\right)\right) \tag{4}$$

where $jsd$ refers to the Jensen-Shannon distance, representing the similarity between two probability distributions. The $RWR_{node}$ refers to the RWR probability vector when one of the nodes of the ego network is selected as seed, and the $RWR_{ego}$ refers to the RWR probability vector when the ego is the seed node. The functional vector is defined as the logarithm of the semantic similarity between the ego and any other nodes in the network (5).

$$functional\,distance = 1000 \times log_2\left(Resnik\left(ego,node\right)\right) \tag{5}$$

where Resnik(ego, node) represents the semantic similarity measure between the ego and the node under consideration. To assess the most similar nodes to the ego, the Kernel Density Estimation (KDE) measure (with Gaussian kernel and bandwidth estimated using the Silvermann formula) to assess the most similar nodes to the ego, is employed. KDE estimates the probability density function (PDF) of the topological and semantic similarity vectors obtained at the previous step. For each ego network, we selected only those nodes within a 0.7≤PDF ≤ 1.0 of both topological and functional similarity. All the nodes overcoming this threshold are selected for the enrichment analysis against the cellular component domain of GO.

## Ontology analyses of extreme deviating proteins and phosphoproteins using PANTHER protein class

We filtered the proteins and phosphosites with an absolute log2 deviation value larger than the 95th percentile (prctile Matlab function) of the distribution of absolute log2 deviation values. Because deviation values were calculated separately for each ventralized genotype (*Toll*[10B]/df or *spn27A*[ex]/df), we obtained two independent lists of proteins and phosphosites with deviation values exceeding

the 95th percentile. From these lists, we selected the proteins and phosphosites whose log2 deviations exceeded the 95th percentile threshold with both ventralized genotypes. For phosphosites, we used the host phosphoprotein for the ontology analyses. When two or more phosphosites with large deviations were hosted by the same phosphoprotein, we counted the phosphoprotein only once. We therefore obtained 206 proteins and 154 phosphoproteins (191 phosphosites) that were used in the ontology analyses.

The ontology analyses were performed using the PANTHER platform (http://www.pantherdb.org/, release PANTHER 17.0 dated February 23rd 2022). We queried the 'Functional classification gene list', based on 'Drosophila melanogaster' organism data. We used the FBgn (Flybase Gene Number) of the proteins and phosphoproteins to produce the query in Panther, and focused on the 'Protein Class' classification. The protein class query allocated 125/206 proteins and 110/154 phosphoproteins to protein class terms. Using gene ontologies from Flybase we manually allocated 48/206 proteins and 35/154 phosphoproteins to one or more of the 24 parental protein class categories. 33/206 proteins and 9/154 phosphoproteins could not be allocated to any parental class category, remained unassigned and were therefore excluded from the reported analyses. In summary, the reported protein class ontology analyses of extremely deviating proteins and phosphoproteins is based on 173/206 proteins and 145/154 phosphoproteins.

## Functional perturbations on microtubules

### Depolymerisation of microtubules and imaging

Embryos laid by flies heterozygous for EMTB-3xGFP and 3xmScarlet-CaaX transgenes or EGFP-CaaX and H2Av-mRFP1 (*Schuh et al., 2007*) transgenes were submerged under Halocarbon oil 27 (Sigma-Aldrich) for staging. Early cellularizing embryos were selected, dechorionated with 50% bleach after removal of Halocarbon oil, washed with $H_2O$, mounted with heptane glue on a coverslip, desiccated with silica gel or Drierite for 10'–15', and subsequently covered with a 3:1 mixture of Halocarbon 700 and 27 oils. For Colcemid injection, 4 mg/ml Demecolcine (Sigma-Aldrich) in $H_2O$ was injected with a custom-made injection needle that was prepared from a borosilicate glass micropipette (Drummond) with a Sutter Instrument pipette puller (P-97/IVF) and a Narishige grinder (EG-44). The stage of injection was controlled based on the transmission brightfield image. A volume of ~65 pL, measured with a 20 X dry lens via an objective micrometre, was injected into the middle section of embryos. Injection was performed with a Narishige IM400 setup mounted on a Nikon Ti2/Eclipse inverted microscope equipped and under a 60 x/NA1.42 oil immersion objective. Imaging was performed on a Yokogawa CSU-W1/SORA imaging system mounted on the same scope. Two laser lines (488 and 561 nm) were used to excite the sample, while a tandem of sCMOS cameras (Prime BSI, Teledyne Photometrics) were used to acquire the image with 2x2 binning. A single z-stack volume was acquired prior to injection, followed by a post-injection z-stack time series. The gap between pre-injection and post-injection imaging was typically 2'~3'. CSU-SORA 4 x zoom was used for imaging EMTB-3xGFP and 3xmScarlet-CaaX with a z-step size of 0.3 μm and a total z-depth of 6.3 μm at a rate of 30 s per volume, while CSU-W1 was used for EGFP-CaaX and H2Av-mRFP1 with a z-step size of 1 μm and a total z-depth of 40 μm at a rate of 1' per volume.

## Image processing and quantitation

For quantitation of nuclear position, single z-slice H2Av-mRFP1 images were converted into tiff format, blurred with a Gaussian filter (σ=2), and segmented with CellPose (v2.2) in 2D using a custom-pretrained nuclear model tailored for each side of the embryo based on manual correction on segmentation generated by a default nuclei model with nuclear diameter set as 20 pixels. The stitch mode was used with a stitch threshold of 0.4 to generate 3D nuclear segments. Nuclear segments were filtered by size (1000–5000 pixels) and height (>10 μm), while those located at the edge of the imaging area were excluded for data processing. The regionprops function implemented in the Skimage Python library was used to define the bounding box of each nuclear segment, from which the middle Z coordinate of the bounding box was designated as the nuclear position. For time alignment, $t_0$ (the onset of gastrulation) was defined as the time point, at which apical constriction in ventral furrow produces a 2.5 μm gap between the cell apex and vitelline membrane for the datasets acquired on the ventral side. Using this $t_0$ designation (from water-injected embryos imaged on the ventral side), the pre-injection cellularization depths were fitted to a linear function based on the assumption that

cellularization depth is linear with time during mid-cellularization. For datasets acquired on the lateral and dorsal sides, the pre-injection timing relative to the onset of gastrulation was derived by plugging in the pre-injection cellularization depth, from which the $t_0$ frame of the dataset was derived. Data processing and plotting were performed with custom-made Python codes using Numpy, Pandas, Matplotlib, and Seaborn libraries.

For *en face* membrane visualization, 3xmScarlet-CaaX images were deconvolved using the Huygens Software (Scientific Volume Imaging) with the deconvolution algorithm Classic MLE using custom parameter sets.

## Acknowledgements

We thank NH Brown, N Bulgakova, S Huelsmann, N Perrimon, V Riechmann, A Stathopoulos, D Stein, S Roth and A Wodarz for reagents and fly stocks, N Lawrence and the Gurdon Institute Imaging Facility for help with 3D-SIM imaging, EMBL Advanced Light Microscopy Facility (ALMF) and CECAD Cologne Imaging Facility for continuous support. Flybase was used throughout the project and is gratefully acknowledged. We also thank Siegfried Roth for critical discussions, and all members of the Leptin lab for discussions throughout the work. This work was supported by funding from the European Molecular Biology Organisation (EMBO), the University of Cologne and the Deutsche Forschungsgemeinschaft (grant DFG LE 546/12).

## Additional information

### Funding

| Funder | Grant reference number | Author |
|---|---|---|
| European Molecular Biology Organization | | Maria Leptin |
| Deutsche Forschungsgemeinschaft | LE 546/12 | Maria Leptin |

The funders had no role in study design, data collection and interpretation, or the decision to submit the work for publication.

### Author contributions

Juan Manuel Gomez, Formal analysis, Supervision, Validation, Investigation, Visualization, Methodology, Writing – original draft, Project administration, Writing – review and editing; Hendrik Nolte, Formal analysis, Validation, Investigation, Writing – review and editing; Elisabeth Vogelsang, Theresa Haunold, Marcus Krüger, Investigation; Bipasha Dey, Michiko Takeda, Investigation, Visualization; Girolamo Giudice, Formal analysis, Visualization, Writing – original draft; Miriam Faxel, Formal analysis, Visualization, Writing – review and editing; Alina Cepraga, Formal analysis; Robert P Zinzen, Formal analysis, Investigation, Methodology, Writing – original draft, Writing – review and editing; Evangelia Petsalaki, Formal analysis, Investigation, Writing – original draft, Writing – review and editing; Yu-Chiun Wang, Formal analysis, Investigation, Visualization, Writing – original draft, Writing – review and editing; Maria Leptin, Conceptualization, Resources, Supervision, Funding acquisition, Visualization, Writing – original draft, Project administration, Writing – review and editing

### Author ORCIDs

Juan Manuel Gomez  http://orcid.org/0000-0002-3041-2503
Hendrik Nolte  http://orcid.org/0000-0003-1560-5099
Elisabeth Vogelsang  https://orcid.org/0000-0002-6817-5953
Girolamo Giudice  https://orcid.org/0000-0002-5359-8208
Theresa Haunold  https://orcid.org/0009-0007-8343-4945
Alina Cepraga  https://orcid.org/0009-0004-6161-1195
Robert P Zinzen  https://orcid.org/0000-0002-8638-5102
Marcus Krüger  https://orcid.org/0000-0002-5846-6941
Evangelia Petsalaki  https://orcid.org/0000-0002-8294-2995
Yu-Chiun Wang  https://orcid.org/0000-0002-3797-4138

Maria Leptin 🔟 https://orcid.org/0000-0001-7097-348X

Decision letter and Author response
Decision letter https://doi.org/10.7554/eLife.99263.sa1
Author response https://doi.org/10.7554/eLife.99263.sa2

## Additional files

### Supplementary files

• Supplementary file 1. Summary of the expression pattern of DV fate markers (dpp, sog, snail) in the wild type and DV patterning mutants.

• Supplementary file 2. Summary the statistical analyses (ANOVA and t-tests) of protein groups and phosphosites that are shown in figures. p values are presented as -log10(p-value).

• Supplementary file 3. Protcome (LFQ) data. p values are presented as -log10(p-value). Empty cell in gene and protein name indicate the detected protein had not been given a gene and/or a protein name in the Uniprot database version used in this study. NaN indicates a protein that was not detected in a particular replicate.

• Supplementary file 4. SILAC Phosphoproteomics data. p values are presented as -log10(p-value). Empty cell in gene and protein name indicate the detected phosphosite is hosted by a protein that had not been given a gene and/or a protein name in the Uniprot database version used in this study. NaN indicates a phosphosite that was not detected in a particular replicate.

• Supplementary file 5. Linear model implementation: IQR values of the deviation distributions obtained with the systematic exploration of dorsal, lateral and ventral domain proportions.

• Supplementary file 6. Log2 deviation values of proteins and phosphosites detected in all genotypes using the dorso-ventral domain proportions: D: 0.4 L: 0.4 V: 0.2. For each experiment (LFQ/Proteomics, SILAC-phosphoproteomics), there is a list of the deviation values of the complete (ANOVA positive and negative) and regulated (ANOVA positive) proteins or phosphosites.

• Supplementary file 7. List of proteins and phosphosites within all DV clusters (1-14). Empty cell in gene and protein name indicate the detected protein (or protein that hosts a phosphosite) had not been given a gene and/or a protein name in the Uniprot database version used in this study.

• Supplementary file 8. RNA Proteome comparison: outcome of the RNA-proteome comparison for the list of genes with a mesoderm label in BDGP (https://insitu.fruitfly.org/cgi-bin/ex/insitu.pl), that are also present in the DV clusters.

• Supplementary file 9. RNA Proteome comparison: list of filtered genes from the RNA atlas (155/8924), with their corresponding: DV RNA reference set, proteome DV cluster, Pearson correlation value that allocated each gene to its DV RNA reference set, and the outcome of the RNA-Proteome comparison. The outcome of the RNA-Proteome comparison (perfect, partial or mismatch) is indicated in two different ways: 1 = 'yes' and 0 = 'no' for each type of outcome, and by coloring the rows (each compared gene-protein distribution); white is perfect overlap, gray is partial overlap and black is a mismatch. Red rows indicate proteins were not assigned to any outcome in the RNA-proteome comparison (Clusters 13 -DLV$_{Tl10B/def}$- and 14 -DLV$_{spn27Aex/def}$-).

• Supplementary file 10. Deviation values of proteins and phosphosites in DV clusters 1-12. Empty cell in gene and protein name indicate the detected protein (or protein that hosts a phosphosite) had not been given a gene and/or a protein name in the Uniprot database version used in this study. NaN indicates it was not possible to calculate the deviation using a particular ventralized mutant.

• Supplementary file 11. Euclidean distance scores of proteins and phosphosites in DV clusters 1-12. Empty cell in gene and protein name indicate the detected protein (or protein that hosts a phosphosite) had not been given a gene and/or a protein name in the Uniprot database version used in this study. NaN indicates it was not possible or did not correspond to calculate the Euclidean distance using a particular ventralized mutant.

• Supplementary file 12. Proteins and phosphoproteins associated with morphogenesis-related cellular components that are significantly-enriched in diffused networks.

• Supplementary file 13. List of proteins and phosphosites with extreme deviations from the linear model.

• MDAR checklist

## Data availability

The whole proteome and phosphoproteomic data is available. The raw files for the proteomics and phosphoproteomics experiments were deposited in PRIDE under separate identifiers: Proteome: Identifier PXD046050 Phosphoproteome: Identifier PXD046192.

The following datasets were generated:

| Author(s) | Year | Dataset title | Dataset URL | Database and Identifier |
|---|---|---|---|---|
| Nolte H, Krüger M | 2024 | Proteome along the dorso-ventral axis of the early Drosophila embryo | https://www.ebi.ac.uk/pride/archive/projects/PXD046050 | PRIDE, PXD046050 |
| Nolte H, Krüger M | 2024 | Phospoproteome along the dorso-ventral axis of the early Drosophila embryo | https://www.ebi.ac.uk/pride/archive/projects/PXD046192 | PRIDE, PXD046192 |

The following previously published dataset was used:

| Author(s) | Year | Dataset title | Dataset URL | Database and Identifier |
|---|---|---|---|---|
| Karaiskos N, Wahle P, Alles J, Boltengagen A | 2017 | Single Cell RNAseq Atlas - *Drosophila* gastrulation | http://www.ncbi.nlm.nih.gov/geo/query/acc.cgi?acc=GSE95025 | NCBI Gene Expression Omnibus, GSE95025 |

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
