## [Editor Report]

This valuable study investigated the changes in the proteome or phosphoproteome during dorsoventral axis specification in the Drosophila embryo. The mutants used to enrich for lateral, ventral, or dorsal regions represent a solid approach, and the large datasets together with subsequent modeling experiments are compelling. This paper provides an excellent resource for the Drosophila community.

---

## [Decision Letter]

[Editors' note: this paper was reviewed by Review Commons.]

Thank you for submitting your article "Differential regulation of the proteome and phosphoproteome along the dorso-ventral axis of the early Drosophila embryo" for consideration by eLife. Your article has been reviewed by 2 peer reviewers at Review Commons, and the evaluation at eLife has been overseen by a Reviewing Editor and Claude Desplan as the Senior Editor.

Based on the previous reviews and the revisions, the manuscript has been improved but there are some remaining issues that need to be addressed, as outlined below:

1. Although the manuscript is easier to read, it is still convoluted for a non-specialist audience, and further clarifications throughout the manuscript are needed.

2. Statistical considerations: statistics have been added as supplementary tables, however they are separated from the main figures, and they should instead be included together with the figures, at least in the legends. At the very least, a description of statistical analysis, what the number of asterisks means, etc. should be included in the figure legends.

3. The observed Toll and Cactus phosphorylation still make little biological sense in terms of the expected effects based on the specific genetic perturbations used. The biological insight based on these data is difficult to distill, despite the provided explanation in the rebuttal (perhaps some of it should be included in the paper). Moreover, why are these examples used as evidence for successful identification of relevant phosphosites?

4. The data presented in Fig. 6 are still confusing because no attempts are made to analyze or present the observed GO categories as functional groups (e.g. by highlighting or some other means). The text refers to some groups but it is difficult to extract this information from the current version of the figure.

5. The suggestion to include information and references for the genes and alleles used in this work early on in the text has not been addressed, but this is important because the choice of the alleles is foundational for the design of the study. Multiple genes and alleles are mentioned in the Introduction and Results on pages 3-4 without references or explaining how specific alleles disrupt gene function. Including this information in the Methods section is not sufficient, and it is critical to explain the rationale for using these specific alleles and combinations and refer to prior work. The authors mention Table 1 but such a table did not appear in the revised manuscript, and Supplemental Table 1 does not provide the relevant information.

6. Some Supplemental Tables (e.g. Supp. Table 7) contain many gene/protein identifiers labeled as "NaN" - it would be worth ensuring that there have been no errors in making these tables.

---

## [Author Response]

Based on the previous reviews and the revisions, the manuscript has been improved but there are some remaining issues that need to be addressed, as outlined below:1. Although the manuscript is easier to read, it is still convoluted for a non-specialist audience, and further clarifications throughout the manuscript are needed.

We have further simplified some of the tricky parts and hope they are more easily digestible now (but recognize they remain difficult).

2. Statistical considerations: statistics have been added as supplementary tables, however they are separated from the main figures, and they should instead be included together with the figures, at least in the legends. At the very least, a description of statistical analysis, what the number of asterisks means, etc. should be included in the figure legends.

We have done this.

3. The observed Toll and Cactus phosphorylation still make little biological sense in terms of the expected effects based on the specific genetic perturbations used. The biological insight based on these data is difficult to distill, despite the provided explanation in the rebuttal (perhaps some of it should be included in the paper). Moreover, why are these examples used as evidence for successful identification of relevant phosphosites?

We agree it would be bold to infer new biological insights on the Toll signalling pathway from our current results, especially because the peak activity of the pathway is an hour before the time point we assay here. At this time point, the transcriptional output, i.e. high expression of twist and snail, repression of zen etc., is fully established, and for all we know, we may be seeing the effects of pathway downregulation or feedback loops rather than signs of primary activity. We have added a comment on this in the main text.

Moreover, why are these examples used as evidence for successful identification of relevant phosphosites?

We used the three Cactus phosphosites S463, S467 and S468, initially identified and characterised by Liu Z.P. et al. (Genes and Development, 1997), and S871 in Toll, identified in Zhai B. et al. (J. Proteome Research, 2008), only to confirm that our approach did find known phosphosites in proteins that are active at this stage (and, incidentally, in the determination of cell fates that are the subject of this study). we have hopefully made this clearer by tweaking the text.

4. The data presented in Fig. 6 are still confusing because no attempts are made to analyze or present the observed GO categories as functional groups (e.g. by highlighting or some other means). The text refers to some groups but it is difficult to extract this information from the current version of the figure.

We have made a number of changes to the figure and added some explanations. Specifically, we have added subdivisions to make the groups more easily recognizable and added some labels and explanations for clarity and transparency.

5. The suggestion to include information and references for the genes and alleles used in this work early on in the text has not been addressed, but this is important because the choice of the alleles is foundational for the design of the study. Multiple genes and alleles are mentioned in the Introduction and Results on pages 3-4 without references or explaining how specific alleles disrupt gene function. Including this information in the Methods section is not sufficient, and it is critical to explain the rationale for using these specific alleles and combinations and refer to prior work. The authors mention Table 1 but such a table did not appear in the revised manuscript, and Supplemental Table 1 does not provide the relevant information.

Indeed, there are only Supplemental Files, (named before as Supplementary Tables) and Supplemental File 1 (named before as Supplementary Table 1) is only an overview of the marker gene expressions in the mutants - sorry for having confused the reader. But as suggested, we have now included the description of and rationale for the allelic combinations we used in the main text.

6. Some Supplemental Tables (e.g. Supp. Table 7) contain many gene/protein identifiers labeled as "NaN" - it would be worth ensuring that there have been no errors in making these tables.

We had used this terminology in the tables of the proteomics experiments to make it clear that missing entries were not errors or forgotten numbers. NaN indicates that proteins were not detected or values could not be calculated. We have now included this explanation in the Supplementary File (named before as Supplementary Table) legends.

What the NaN signified in the columns with the gene and protein names is that the protein or gene had not been given a gene name in the Uniprot database at the time. But we agree it does not make so much sense to use this terminology here, so we have now left the spaces blank.

The number of missing gene and protein names may seem high, but this is what the database showed at the time. Any names that have been assigned (or changed) since then can of course be found easily via the Uniprot number.